# Structural insights into mRNA reading frame regulation by tRNA modification and slippery codon–anticodon pairing

Eric D Hoffer[1], Samuel Hong[1], S Sunita[1], Tatsuya Maehigashi[1], Ruben L Gonzalez Jnr[2], Paul C Whitford[3], Christine M Dunham[1]*

[1]Department of Biochemistry, Emory University School of Medicine, Atlanta, United States; [2]Department of Chemistry, Columbia University, New York, United States; [3]Department of Physics, Northeastern University, Boston, United States

**Abstract** Modifications in the tRNA anticodon loop, adjacent to the three-nucleotide anticodon, influence translation fidelity by stabilizing the tRNA to allow for accurate reading of the mRNA genetic code. One example is the N1-methylguanosine modification at guanine nucleotide 37 ($m^1G37$) located in the anticodon loop and immediately adjacent to the anticodon nucleotides 34, 35, 36. The absence of $m^1G37$ in $tRNA^{Pro}$ causes +1 frameshifting on polynucleotide, slippery codons. Here, we report structures of the bacterial ribosome containing $tRNA^{Pro}$ bound to either cognate or slippery codons to determine how the $m^1G37$ modification prevents mRNA frameshifting. The structures reveal that certain codon–anticodon contexts and the lack of $m^1G37$ destabilize interactions of $tRNA^{Pro}$ with the P site of the ribosome, causing large conformational changes typically only seen during EF-G-mediated translocation of the mRNA-tRNA pairs. These studies provide molecular insights into how $m^1G37$ stabilizes the interactions of $tRNA^{Pro}$ with the ribosome in the context of a slippery mRNA codon.

*For correspondence:
cmdunha@emory.edu

Competing interests: The authors declare that no competing interests exist.

## Introduction

Post-transcriptionally modified RNAs, including ribosomal RNA (rRNA), transfer RNA (tRNA) and messenger RNA (mRNA), stabilize RNA tertiary structures during ribonucleoprotein biogenesis, regulate mRNA metabolism, and influence other facets of gene expression. RNA modifications located in the three-nucleotide anticodon of tRNAs contribute to accurate protein synthesis and expand the tRNA coding capacity by facilitating Watson–Crick-like base-pairs with mRNA codons. The most commonly modified tRNA anticodon position is at nucleotide 34 and the modifications to this nucleotide are functionally important for gene expression by facilitating base-pairing interactions with the mRNA codon (*Agris et al., 2017*; *Boccaletto et al., 2018*; *Agris et al., 2018*). Other tRNA nucleotides also contain highly conserved nucleotide modifications, but their precise roles in protein synthesis are not fully understood. For example, nucleotide 37 is adjacent to the anticodon (*Figure 1A*) and is modified in >70% of all tRNAs (*Machnicka et al., 2014*). The two most common modifications at nucleotide 37 are 6-threonylcarbamoyladenosine ($t^6A$) and 1-methylguanosine ($m^1G$), which together account for >60% of nucleotide 37 modifications (*Boccaletto et al., 2018*). Both the $t^6A37$ modification in $tRNA^{Lys}$ and the $m^1G37$ modification in $tRNA^{Pro}$ are thought to stabilize the tertiary structure of the anticodon loop for high-affinity binding to their cognate codons (*Murphy et al., 2004*; *Sundaram et al., 2000*; *Nguyen et al., 2019*). The $m^1G37$ modification also prevents the ribosome from shifting out of the three-nucleotide mRNA codon frame ('frameshifting') to ensure the correct polypeptide is expressed (*Björk et al., 1989*; *Li et al., 1997*; *Urbonavicius et al., 2001*; *Gamper et al., 2015a*). However, the molecular basis for how the $m^1G37$ modification maintains the mRNA reading frame is unknown.

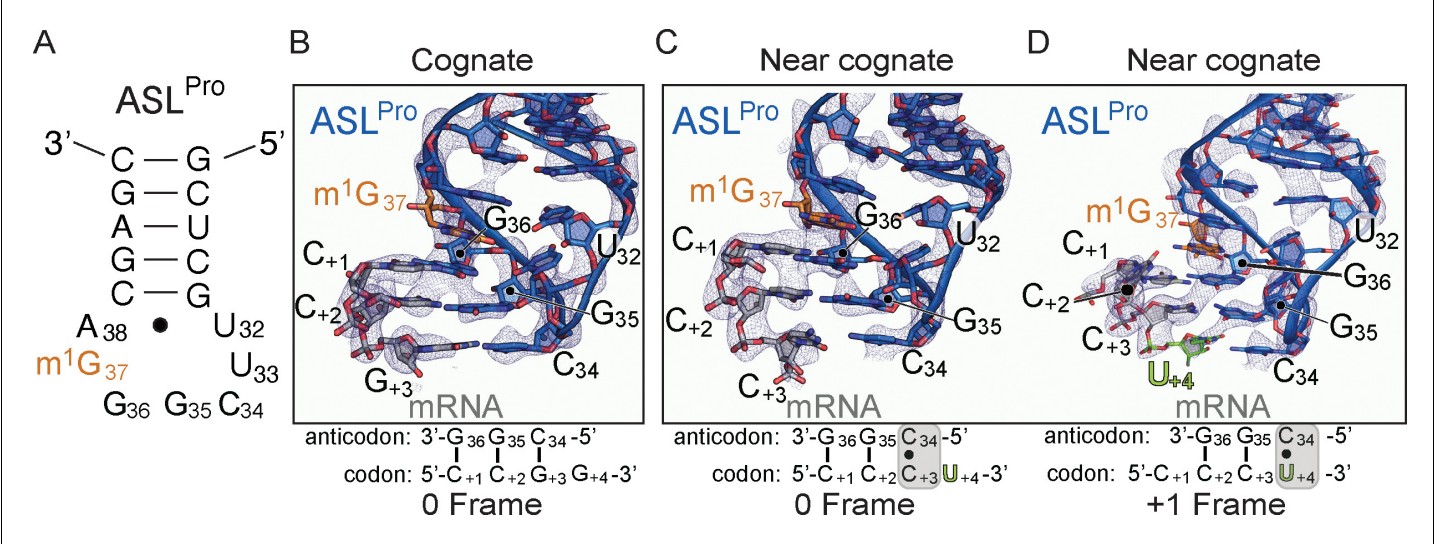

**Figure 1.** The ability of ASL$^{Pro}$ to +1 frameshift in the peptidyl (P) site is dependent on the near-cognate mRNA codon. (**A**) Secondary structure of the anticodon stem-loop (ASL) of tRNA$^{Pro}$. The m$^1$G37 modified nucleotide is shown in orange and the anticodon nucleotides (C34, G35, and G36) and the U32-A38 pairing are labeled. (**B**) Structure of 70S-ASL$^{Pro}$ bound to a cognate CCG codon in the P site with the codon in the 0 or canonical frame. All 2F$_o$-F$_c$ electron density maps shown in panels B-D are contoured at 1.0σ. (**C,D**) Structure of 70S-ASL$^{Pro}$ bound to a +1 slippery CCC-U codon shows the mRNA position is either in the 0 (panel C) or +1 frame (panel D) in the two 70S molecules in the crystallographic asymmetric unit. In either the 0 or +1 frame, a *cis* Watson–Crick interaction at the third base-pair forms (C$_{+3}$•C34 or U$_{+4}$•C34). mRNA numbering starts at +1 according to the first position in the P site.

The online version of this article includes the following figure supplement(s) for figure 1:

**Figure supplement 1.** Determining the mRNA frame by visualizing the +4 nucleotide mRNA phosphate density.

The m$^1$G37 modification is present in >95% of all three tRNA$^{Pro}$ isoacceptors across all three domains of life (*Boccaletto et al., 2018*; *Björk et al., 1989*). The absence of m$^1$G37 in tRNA$^{Pro}$ causes high levels of +1 frameshifting on so-called 'slippery' or polynucleotide mRNA sequences where four nucleotides encode for a single proline codon (*Björk et al., 1989*). In the case of the tRNA$^{Pro}$-CGG isoacceptor (anticodon is shown 5'-3'), a +1 slippery codon consists of the CCC proline codon and either an additional C or U to form a four-nucleotide codon, CCC-(U/C) (*Sroga et al., 1992*; *Qian et al., 1998*) (codons depicted 5'-3'). tRNA$^{Pro}$-CGG lacking m$^1$G37 results in the ribosome being unable to distinguish a correct from an incorrect codon–anticodon interaction during decoding at the aminoacyl (A) site (*Nguyen et al., 2019*; *Maehigashi et al., 2014*). However, this miscoding event does not cause the shift in the mRNA frame in the A site despite the tRNA$^{Pro}$-CGG anticodon stem-loop nucleotides become more mobile (*Maehigashi et al., 2014*). A post-decoding frameshift event is further supported by detailed kinetic analyses (*Gamper et al., 2015a*). The absence of the m$^1$G37 modification in tRNA$^{Pro}$ causes a ~ 5% frameshifting frequency during translocation of the mRNA-tRNA pair from the A to the peptidyl (P) site and a ~40% frameshifting frequency once the mRNA-tRNA pair has reached the P site. When tRNA$^{Pro}$-CGG lacks m$^1$G37 and decodes a +1 slippery codon, both the process of translocation and the unique environment of the P site appear to contribute to the inability of the ribosome to maintain the mRNA frame.

In circumstances where mRNA frameshifting is caused by changes in the tRNA such as the absence of modifications or changes in the size of the anticodon loop as found in frameshift suppressor tRNAs, primer extension assays demonstrated that the shift into the +1 frame is observed upon direct tRNA binding at the P site (*Phelps et al., 2006*; *Walker and Fredrick, 2006*). These studies show that the nature of the interactions between the frameshift-prone mRNA-tRNA pair and the ribosomal P site directly permit frameshifting. Furthermore, the presence of a nascent polypeptide chain, the acylation status of the tRNA (deacylated or aminoacylated), and even the presence of a tRNA in the A site do not appear to contribute to the ability of these tRNAs to cause frameshifting. Structural studies of frameshift-prone or suppressor tRNAs have provided molecular insights into how such tRNAs may dysregulate mRNA frame maintenance (*Maehigashi et al., 2014*; *Fagan et al.,*

*2014*; *Hong et al., 2018*). Frameshift suppressor tRNA[SufA6] contains an extra nucleotide in its anticodon loop and undergoes high levels of +1 frameshifting on slippery proline codons (*Björk et al., 1989*; *Qian et al., 1998*). The structure of the anticodon stem-loop (ASL) of the tRNA[SufA6] bound directly at the P site revealed that its anticodon engages the slippery CCC-U proline codon in the +1 frame (*Hong et al., 2018*). Further, the full-length tRNA[SufA6] bound at the P site reveals that the small 30S subunit head domain swivels and tilts, a movement that is similar to the one that is caused by the GTPase elongation factor-G (EF-G) upon translocation of the tRNAs through the ribosome that several groups have attempted to characterize (*Ermolenko and Noller, 2011*; *Wasserman et al., 2016*; *Belardinelli et al., 2016a*, *Nguyen and Whitford, 2016*, *Guo and Noller, 2012*). The process of mRNA-tRNA translocation is coupled to this head domain swivel and tilting which is distinct from other conformational changes the ribosome undergoes including intersubunit rotation (*Wasserman et al., 2016*; *Ratje et al., 2010*; *Zhou et al., 2013*; *Zhou et al., 2014*; *Holtkamp et al., 2014*; *Belardinelli et al., 2016b*; *Guo and Noller, 2012*, *Nguyen and Whitford, 2016*). However, the 70S-tRNA[SufA6] structure is in the absence of EF-G, suggesting that a +1 frameshift event caused by frameshift-prone tRNAs dysregulates some aspect of translocation. Although these studies demonstrate that +1 frameshift-prone tRNAs are good model systems to uncover mechanisms by which both the ribosome and the mRNA-tRNA pair contribute to mRNA frame maintenance, it is unclear whether normal tRNAs lacking modifications result in +1 frameshifting in the same manner. To address this question, here we solved six 70S structures of tRNA[Pro]-CGG in the presence or absence of the m[1]G37 modification and bound to either cognate or slippery codons in the P site. Our results define how the m[1]G37 modification stabilizes the interactions of tRNA[Pro]-CGG with the ribosome. We further show that ribosomes bound to mRNA-tRNA pairs that result in +1 frameshifts promoted by tRNA[Pro]-CGG result in large conformational changes of the 30S head domain, consequently biasing the tRNA-mRNA pair toward the E site.

## Results

### A near-cognate interaction between ASL[Pro] and the slippery CCC-U proline codon *alone* causes a shift into the +1 frame

To address whether tRNA[Pro]-CGG induces a +1 frameshift by a similar mechanism as tRNA[SufA6], we determined two X-ray structures of ASL[Pro] decoding either a cognate CCG or a near-cognate, slippery CCC-U codon bound at the P site (*Figure 1* and *Tables 1* and *2*). We used chemically synthesized ASL[Pro] (17 nucleotides) to ensure G37 is fully methylated at the N1 position (*Figure 1A*). The structure of P-site ASL[Pro] interacting with a cognate CCG proline codon was solved to a resolution of 3.1 Å and reveals that the three nucleotides of the anticodon (nucleotides G36-G35-C34) form three Watson–Crick base-pairs with the $C_{+1}$-$C_{+2}$-$G_{+3}$ mRNA codon, respectively (*Figure 1A*) (mRNA nucleotides are numbered starting at +1 from the P-site codon). The anticodon interacts with the codon in the canonical or 0 mRNA frame indicating a frameshift has not occurred.

We next asked how ASL[Pro] interacts in the P site with a CCC-U codon, a slippery codon known to facilitate +1 frameshifts (*Björk et al., 1989*). In this X-ray structure solved to a resolution of 3.4 Å, we find two different conformations of the codon–anticodon interaction in the two molecules of the crystallographic asymmetric unit (*Figure 1C and D* and *Table 2* and *Video 1*). Although it is common for one 70S ribosome to contain better electron density than the other in the asymmetric unit in this particular crystal form (*Selmer et al., 2006*), to our knowledge, it is not common to obtain two different conformational states of the same functional complex. In the first ribosome molecule, nucleotides G36 and G35 of ASL[Pro] form Watson–Crick base-pairs with the first two nucleotides of the 0-

**Table 1.** RNAs used in this study.

| | |
|---|---|
| tRNA[Pro] 5' half | 5'-CGGUGAUUGGCGCAGCCUGGUAGCGCACUUCGUUCGGm[1]GA-3' |
| tRNA[Pro] 3' half | 5'-CGAAGGGGUCGGAGGUUCGAAUCCUCUAUCACCGACCA-3' |
| mRNA_cognate | 5'- GGCAAGGAGGUAAAA CCGG-3' |
| mRNA_slippery | 5'- GGCAAGGAGGUAAAA CCCU-3' |

The underlined nucleotides indicates the Shine-Dalgarno region while the bold nucleotides are the P-site codons.

**Table 2.** Data collection and refinement statistics for 70S-ASL$^{Pro}$ structures.

| | 70S-ASL$^{Pro}$-CCG codon | 70S-ASL$^{Pro}$-CCC-U codon |
|---|---|---|
| Data collection | | |
| Space group | $P2_12_12_1$ | $P2_12_12_1$ |
| Wavelength (Å) | 0.9791 | 0.9791 |
| Cell dimensions | | |
| a, b, c (Å) | 209.79,451.91,621.58 | 210.12,451.80,622.96 |
| α, β, γ (°) | 90, 90, 90 | 90, 90, 90 |
| Resolution (Å) | 49.70–3.10 (3.21–3.10) | 49.70–3.40 (3.52–3.40) |
| $R_{pim}$ (%) | 21.2 (80.5) | 14.5 (65.5) |
| I/σI | 4.7 (1.1) | 5.9 (1.3) |
| Completeness (%) | 97.99 (87.41) | 98.85 (94.87) |
| Redundancy | 3.7 (3.1) | 4.2 (3.4) |
| CC1/2 | 0.968 (0.250) | 0.987 (0.359) |
| Refinement | | |
| Reflections | 1034968 (91757) | 797600 (75973) |
| $R_{work}/R_{free}$ (%) | 24.1/27.2 | 19.8/23.7 |
| No. atoms | 289313 | 290047 |
| B-factors (Å$^2$) | | |
| Overall | 90.37 | 103.79 |
| Macromolecule | 90.6 | 104.05 |
| Ligand/ion | 39.43 | 43.84 |
| R.m.s deviations | | |
| Bond lengths (Å) | 0.004 | 0.010 |
| Bond angles (°) | 0.83 | 0.98 |
| PDB ID | 6NTA | 6NSH |

Highest resolution shell is shown in parentheses.

frame mRNA codon (*Figure 1C*). A mismatch C$_{+3}$•C34 forms at the third base-pair position of the codon–anticodon interaction and contains a single hydrogen bond between the Watson–Crick face of C$_{+3}$ (the N3 position) and C34 (the N4 position). This codon–anticodon interaction is thus defined as near-cognate because of the single mismatch. The distance between the anticodon nucleotide C34 and the mRNA nucleotide C$_{+3}$ is increased in the P site as compared to this same mRNA-tRNA pair in the A site (3.6 Å vs. 3.1 Å *Maehigashi et al., 2014*), indicating that the interaction has weakened. Although this interaction is at the distance limit of a hydrogen bond, C34 and C$_{+3}$ are positioned to form a *cis*-Watson–Crick pair, similar to the orientation observed in the A site (*Maehigashi et al., 2014*; *Figure 1—figure supplement 1C*). The codon–anticodon interaction is in the 0 frame as indicated by the clear phosphate density of the next mRNA nucleotide, U$_{+4}$ (*Figure 1—figure supplement 1C*). The mRNA used in these studies only contained a single nucleotide after the three-nucleotide codon programmed in the P site to allow for the unambiguous identification of the reading frame.

Strikingly, in the other ribosome molecule in the crystallographic asymmetric unit, the codon–anticodon interaction is in the +1 frame (*Figure 1D* and *Video 1*). The first two nucleotides of the anticodon (G36-G35) form Watson–Crick base-pairs with C$_{+2}$-C$_{+3}$ nucleotides of the mRNA codon, respectively. The first nucleotide of the proline codon, C$_{+1}$, no longer interacts with the tRNA and does not appear to make any interactions with the ribosome. At the third position of the codon–anticodon interaction, a *cis*-Watson–Crick U$_{+4}$•C34 pair forms that contains a single hydrogen bond between the Watson–Crick face of U$_{+4}$ (the O4 position) and C34 (the N4 position) (*Figure 1—figure supplement 1D*). The absence of electron density for the next nucleotide in the mRNA confirms

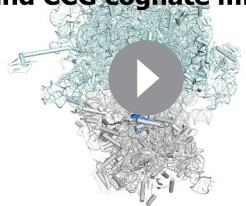

**70S complexed with ASLPro and CCG cognate mRNA**

**Video 1.** mRNA +1 frameshifting of ASL<sup>Pro</sup> in the peptidyl (P) site is dependent on the slippery mRNA codon. An overview of the 70S ribosome (50S is light cyan and the 30S in gray) showing how the anticodon stem-loop (ASL) of tRNA<sup>Pro</sup> (blue) interacts with either a cognate CCG codon or a slippery CCC-U codon (black). The m$^1$G37 modified nucleotide is shown in orange. A zoomed-in view of the anticodon nucleotides (G36, G35, and C34) interacting with the cognate C$_{+1}$, C$_{+2}$ and G$_{+3}$ mRNA respectively, in the 0 frame. All 2F$_o$-F$_c$ electron density maps shown are contoured at 1.0 σ. A morph from the 0 frame structure to the structure of ASL<sup>Pro</sup> interacting with a slippery CCC-U codon is shown to demonstrate the movement of the mRNA into the +1 frame. The fourth nucleotide of the mRNA codon is shown in green. In either the 0 or +1 frame, a *cis* Watson–Crick interaction at the third base-pair forms (C$_{+3}$•C34 or U$_{+4}$•C34). mRNA numbering starts at +1 according to the first position of the mRNA in the P site.

https://elifesciences.org/articles/51898#video1

the +1 frame of the codon–anticodon interaction as this is the last nucleotide of the mRNA (*Figure 1—figure supplement 1D*). Therefore, although the m$^1$G37 modification is present in ASL<sup>Pro</sup>, the shift in the +1 frame can still occur. These data indicate that the presence of a near-cognate, slippery codon is sufficient to promote a +1 mRNA frameshift. Consistent with our observations, biochemical studies of other near-cognate mRNA-tRNA pairs in the P site also show frameshifting in both directions on the mRNA (*Zaher and Green, 2009*).

## The absence of m$^1$G37 in tRNA$^{Pro}$-CGG bound to a cognate CCG codon does not destabilize its interactions with the ribosome

We next asked how full-length tRNA<sup>Pro</sup>-CGG bound at the P site interacts with mRNA. Our recent structure of the ribosome with +1 frame-shift suppressor tRNA<sup>SufA6</sup> indicated that although the tRNA was placed at the P site, when tRNA<sup>SufA6</sup> engaged with mRNA that causes +1 frameshifts, the tRNA moves towards the E site and induces large conformational changes of the 30S subunit, specifically the head domain (*Hong et al., 2018*). We wanted to directly compare these structures given that the +1 frameshift is induced by different signals: an extra nucleotide in the anticodon loop in tRNA<sup>SufA6</sup> and the absence of m$^1$G37 in tRNA<sup>Pro</sup>-CGG. It is possible that tRNA<sup>SufA6</sup> causes destabilization and conformational changes of the 30S head domain

because of the extra nucleotide in its ASL; such a mechanism would not apply for m$^1$G37 in tRNA<sup>Pro</sup>-CGG. As a control, we solved a structure of tRNA<sup>Pro</sup>-CGG containing the m$^1$G37 modification bound to a cognate CCG codon in the ribosomal P site to a resolution of 3.2 Å (*Figure 2A* and *Table 3* and *Video 2*). tRNA<sup>Pro</sup>-CGG binds in the classical P/P state (bound to the P site on the 50S and the 30S subunits) and interacts with the proline codon in the 0 frame. Specifically, the G36-G35-C34 anticodon nucleotides form three Watson–Crick base-pairs with the C$_{+1}$-C$_{+2}$-G$_{+3}$ proline codon nucleotides, respectively (*Figure 2E*, *Figure 2—figure supplement 1A and B*). The methyl group at position 1 of the nucleobase of G37 stacks with the nucleobase of anticodon nucleotide G36 to form a canonical four-nucleotide stack between nucleotides 34–37. This anticodon stack is important for productive interactions with both mRNA and the ribosome (*Maehigashi et al., 2014*; *Grosjean et al., 1976*; *Gustilo et al., 2008*). The P/P location of tRNA<sup>Pro</sup>-CGG is thus consistent with functional assays demonstrating that tRNA<sup>Pro</sup> isoacceptors do not appear to spontaneously move into the +1 frame on non-slippery codons (*Björk et al., 1989*; *Gamper et al., 2015a*).

We next solved a 3.9 Å structure of tRNA<sup>Pro</sup>-CGG lacking the m$^1$G37 modification (G37 Δm$^1$) and interacting with a cognate CCG proline codon in the P site (*Figure 2B*, *Figure 2—figure supplement 1C and D* and *Table 3* and *Video 2*). Although the m$^1$G37 modification was previously shown to be critical for high-affinity binding and decoding of the cognate CCG codon at the A site (*Nguyen et al., 2019*), its influence on the overall conformation of tRNA<sup>Pro</sup>-CGG in the P site appears to be minimal as the tRNA adopts a P/P orientation (*Figure 2B*). Notably, the mRNA remains in the 0 frame with three Watson–Crick base-pair interactions between the codon and the anticodon (*Figure 2F*). Similar to the structural studies of the ASL<sup>Pro</sup> in the 0 frame (*Figure 1B and D*), the mRNA used in these studies contains an additional nucleotide after the P-site CCG codon (*Figure 1*; *Figure 2B* and *Table 1*). The mRNA has clear phosphate density for the single A-site

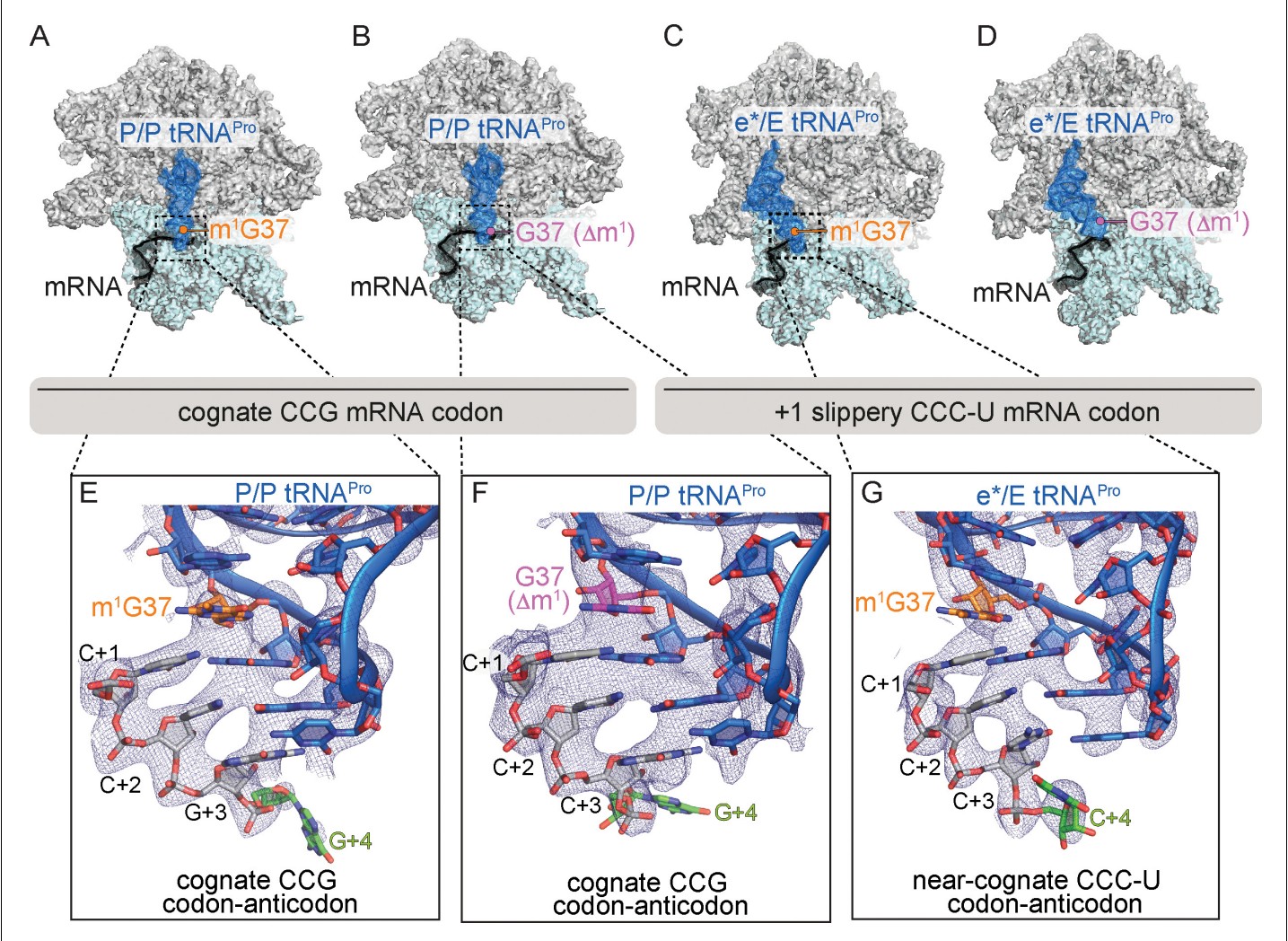

**Figure 2.** Identity of the mRNA proline codon regulates 30S head domain swivel and tilting. Overview of 70S ribosome-tRNA[Pro] complex structures: (**A**) tRNA[Pro] m1G37 on a cognate CCG codon adopts a P/P orientation (located on the P site on the 30S and 50S subunits); (**B**) tRNA[Pro] lacking the m1G37 modification (G37 Δm1) on a cognate CCG codon also adopts a P/P orientation; (**C**) tRNA[Pro] on a +1 slippery CCC-U codon adopts an e*/E orientation (e* denotes the location between the E and P sites on the 30S while "E" is the E site of the 50S); and (**D**) tRNA[Pro] lacking the m1G37 (G37 Δm1) on a +1 slippery CCC-U codon adopts an e*/E orientation. In this complex (panel D), the 30S head domain and anticodon-codon interaction are disordered. In panels A-D, the 16S rRNA of the 30S head domain is removed for clarity. Zoomed-in view of 2F$_o$-F$_c$ density of (**E**) the codon–anticodon for tRNA[Pro] on a cognate CCG codon, (**F**) tRNA[Pro] G37 (Δm1) on a cognate CCG codon in the P site, and (**G**) tRNA[Pro] G37 (Δm1) on a near-cognate CCC-U codon in position between the E and the P sites (e*). All 2F$_o$-F$_c$ electron density maps shown in panels E-G are contoured at 1.0σ.

The online version of this article includes the following figure supplement(s) for figure 2:

**Figure supplement 1.** mRNA and full-length tRNA[Pro] electron density.

**Figure supplement 2.** Full-length tRNA[Pro] interactions with 16S nucleotides G1338 and A1339.

**Figure supplement 3.** tRNA[Pro] G37 (Δm1) coupled with a slippery codon capable of +1 frameshifting induces disorder in the 30S head domain.

nucleotide indicating the mRNA remains in the 0 frame (*Figure 2—figure supplement 1D*). The absence of the m1G37 modification correlates with a slightly weaker U32-A38 pairing when compared to modified tRNA[Pro]-CGG (*Figure 2—figure supplement 2B*). These structures suggest that the absence of 1-methyl modification on G37 alone is not sufficient to destabilize the tRNA in the P site of the ribosome. This observation is also consistent with toeprint analyses demonstrating that tRNA[Pro]-GGG lacking m1G37 on a non-slippery codon remained in the 0 frame (*Gamper et al., 2015a*).

**Table 3.** Data collection and refinement statistics for 70S-tRNA$^{Pro}$ structures.
Highest resolution shell is shown in parentheses.

| | tRNA$^{Pro}$ m$^1$G37-CCG codon | tRNA$^{Pro}$ m$^1$G37-CCC-U codon | tRNA$^{Pro}$ G37(Δm$^1$)-CCG codon | tRNA$^{Pro}$ G37(Δm$^1$)-CCC-U codon |
|---|---|---|---|---|
| **Data collection** | | | | |
| Space group | P2$_1$2$_1$2$_1$ | P2$_1$2$_1$2$_1$ | P2$_1$2$_1$2$_1$ | P2$_1$2$_1$2$_1$ |
| Wavelength (Å) | 0.9792 | 0.9792 | 0.9792 | 0.9792 |
| Cell dimensions | | | | |
| a, b, c (Å) | 210.20,451.47,620.21 | 210.74,450.26,626.11 | 209.97,450.71,619.40 | 210.09,450.32,622.89 |
| α, β, γ (°) | 90, 90, 90 | 90, 90, 90 | 90, 90, 90 | 90, 90, 90 |
| Resolution (Å) | 49.20–3.20 (3.31–3.20) | 49.93–3.50 (3.63–3.50) | 49.82–3.97 (4.11–3.97) | 49.83–4.14 (4.29–4.14) |
| R$_{pim}$ (%) | 11.5 (51.7) | 9.00 (44.1) | 8.60 (86.0) | 9.807 (95.5) |
| I/σI | 6.7 (1.5) | 7.8 (1.7) | 6.3 (1.0) | 5.1 (1.0) |
| Completeness (%) | 99.11 (98.45) | 97.55 (89.72) | 98.60 (95.72) | 98.39 (95.63) |
| Redundancy | 5.9 (4.4) | 4.2 (2.2) | 14.1 (10.0) | 6.5 (3.6) |
| CC1/2 | 0.991 (0.377) | 0.996 (0.464) | 0.998 (0.313) | 0.998 (0.37) |
| | | | | |
| **Refinement** | | | | |
| Reflections | 951115 (93932) | 723555 (66052) | 4959167 (47742) | 439195 (42360) |
| R$_{work}$/R$_{free}$ (%) | 22.8/25.6 | 23.4/25.6 | 22.8/25.5 | 24.8/29.4 |
| No. atoms | 291966 | 292039 | 291793 | 291185 |
| B-factors (Å$^2$) | | | | |
| Overall | 103.87 | 113.6 | 177.39 | 247.09 |
| Macromolecule | 104.12 | 113.9 | 177.77 | 247.54 |
| Ligand/ion | 42.04 | 37.71 | 72.42 | 123.12 |
| R.m.s deviations | | | | |
| Bond lengths (Å) | 0.011 | 0.010 | 0.007 | 0.007 |
| Bond angles (°) | 1.00 | 1.23 | 0.94 | 1.38 |
| PDB ID | 6NUO | 6NWY | 6O3M | 6OSI |

## A CCC-U slippery proline codon in the P site causes tRNA$^{Pro}$-CGG destabilization and 30S head movement

In an attempt to reconcile the exact role of the m$^1$G37 modification in tRNA$^{Pro}$-CGG in the context of a +1 slippery CCC-U codon, we next solved a ribosome structure of this complex in the P site to a resolution of 3.2 Å (*Figure 2C* and *Table 3* and *Video 2*). Unexpectedly, tRNA$^{Pro}$-CGG moves from the P site towards the E site, adopting a position on the ribosome which we refer to as e*/E (*Hong et al., 2018*) (where e* signifies an intermediate position between the 30S subunit P and the E sites and E signifies a 50S E-site position) (*Figure 2C*, *Figure 2—figure supplement 1E*). The e*/E position of tRNA$^{Pro}$-CGG when bound to this slippery codon is different from the classical P/P position that the same tRNA adopts when it interacts with a cognate CCG codon (compare *Figure 2A* with *Figure 2C*). The classical P/P tRNA$^{Pro}$-CGG location on its cognate codon is consistent with dozens of other P-site tRNA bound ribosome structures. Therefore, the ability of tRNA$^{Pro}$-CGG to move towards the E site when bound to a slippery proline codon suggests that it is this near-cognate codon–anticodon interaction alone that is sufficient to destabilize tRNA$^{Pro}$-CGG (*Figure 2C*).

The e*/E position of the tRNA$^{Pro}$-CGG-mRNA pair is additionally coupled to large conformational changes of the 30S head domain (*Figure 3* and *Video 2*). 30S head motions accompanying canonical translocation or caused by a frameshift-prone tRNAs have been previously described by our group

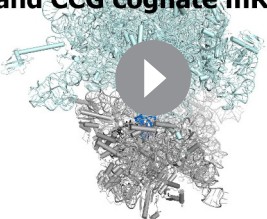

**70S complexed with tRNAPro and CCG cognate mRNA**

**Video 2.** Influence of m$^1$G37 and the slippery codon on +1 frameshifting and conformational changes of the 30S head domain. An overview of the 70S ribosome (50S is light cyan and the 30S in gray) showing how tRNA$^{Pro}$ (blue) (+/- m$^1$G37) interacts with either a cognate CCG codon or a slippery CCC-U codon (black). A zoomed-in view of the anticodon nucleotides (G36, G35, and C34) interacting with the cognate C$_{+1}$, C$_{+2}$, and G$_{+3}$ mRNA respectively, in the 0 frame. All 2F$_o$-F$_c$ electron density maps shown are contoured at 1.0σ. The m$^1$G37 modified nucleotide is shown in orange and the U32-A38 pairing is indicated. A morph of the changes of the U32-A38 pairing when the tRNA interacts with either a cognate CCG or a slippery CCC-U codon is shown. An overview and a morph of P/P tRNA$^{Pro}$ bound to a cognate CCG codon to the e*/E site along with movement of the 30S head domain is shown. Lastly, a zoomed-in view shows a morph from a 0 frame codon-anticodon interaction that forms in the P/P site to the pairing in the e*/E site.
https://elifesciences.org/articles/51898#video2

and others (*Hong et al., 2018*; *Ratje et al., 2010*; *Zhou et al., 2013*; *Zhou et al., 2014*; *Mohan et al., 2014*; *Zhou et al., 2019*), but the two distinct movements of "swiveling" and "tilting" have never been analyzed separately (*Figure 3*). To more precisely describe this multidimensional motion, we used previously reported procedures to define 'swivel' as the movement of the head relative to the body within a plane (i.e. pure rotation about a single axis defined by structures of the unrotated and swiveled conformations (using PDB codes 4V9D and 4V4Q, respectively)), while 'tilt' describes any deviations from pure rotation (*Nguyen and Whitford, 2016*). Together, these more accurate calculations provide an unambiguous description of the full, three-dimensional orientation of the head during translocation, where the tRNAs are found in intermediate states (*Zhou et al., 2013*; *Zhou et al., 2014*). The head domain swivels in a ~18° counterclockwise direction while also undergoing a ~5° tilt away from the ribosome and perpendicular to the mRNA path (*Ratje et al., 2010*; *Zhou et al., 2013*; *Zhou et al., 2014*) (the counterclockwise swivel is defined of the ribosome viewed with the E-, P-, A-tRNA sites oriented from left to right as depicted in *Figure 2*). The head domain is comprised of 16S rRNA nucleotides 930–1380 and seven ribosomal proteins (S3, S7, S9, S10, S13, S14, and S19) (*Belardinelli et al., 2016a*; *Guo and Noller, 2012*) that collectively move as a rigid body during EF-G-mediated translocation of the mRNA-

tRNA pairs on the 30S (*Ratje et al., 2010*; *Zhou et al., 2013*; *Zhou et al., 2014*). The same swivel and tilting of the head domain of e*/E tRNA$^{Pro}$-CGG occurs but our structure lacks EF-G. This result indicates that even with the m$^1$G37 modification, interactions of tRNA$^{Pro}$-CGG with the P site are destabilized when bound to a near-cognate codon, promoting its movement toward the E site along with 30S head domain swivel and tilting.

## The mRNA located in the 30S E and P sites is constricted when tRNA$^{Pro}$-CGG adopts an e*/E position

The path of the mRNA on the ribosome accommodates three nucleotides, or a single codon, in each of the A, P, and E sites (*Figure 4A and B*). The ribosome interacts extensively with the mRNA in the A site to ensure accurate decoding but interacts less substantially with the P-site codon, while the E-site mRNA codon is the least monitored. The ribosome interacts with E-site mRNA at two positions through non-sequence-specific contacts: 16S rRNA nucleotide G693 stacks with the first nucleotide of the E-site mRNA codon and the G926 nucleobase contacts the phosphate of the third mRNA codon nucleotide (*Figure 4B and C*; *Selmer et al., 2006*). The e* position of tRNA$^{Pro}$-CGG signifies that the tRNA is closer to the E site than the P site on the 30S, however it is similar to the pe/E tRNA position seen in translocation intermediate ribosome structures containing EF-G (*Figure 4D*; *Zhou et al., 2014*). Only two nucleotides of the codon–anticodon interaction of the pe/E tRNA are positioned in the E site leaving a pocket for an additional nucleotide to occupy upon full translocation of the mRNA-tRNA pair (*Zhou et al., 2013*; *Zhou et al., 2014*) (compare *Figure 4C* with *Figure 4D*). In our structure containing an e*/E tRNA$^{Pro}$-CGG bound to a near-cognate, slippery codon, the first three nucleotides of the codon fully occupy the E site (*Figure 4E*), despite the codon–anticodon adopting an intermediate position on the 30S between the P and the E sites. This

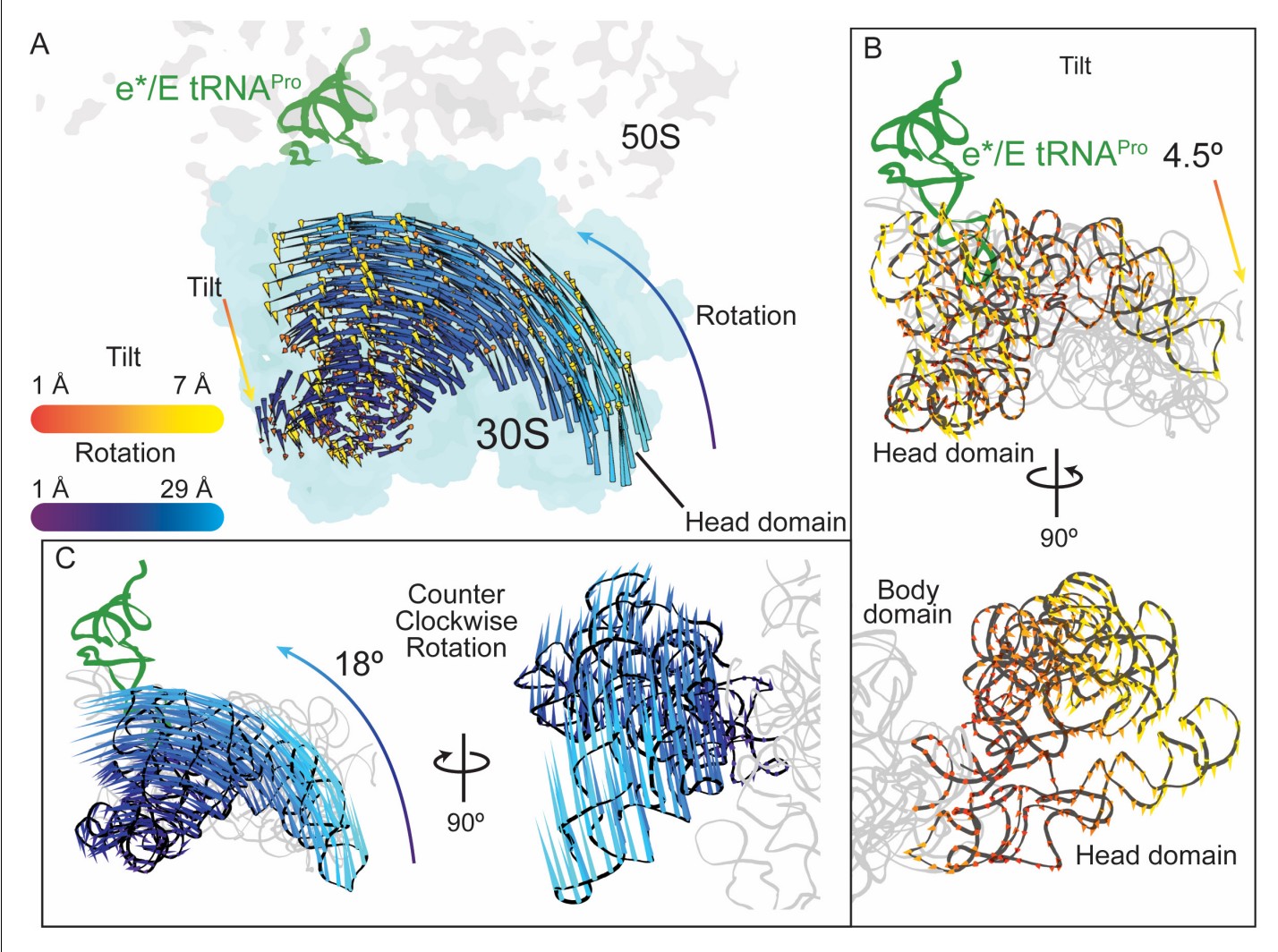

**Figure 3.** 30S head domain movement in the presence of a +1 frameshift-prone tRNA. (**A**) Overview of the 70S ribosome containing an e*/E tRNA^Pro bound to a +1 slippery CCC-U codon. Shifts in phosphate atom positions of the 30S head domain (16S rRNA nucleotides 930–1380) in this structure as compared to the unrotated 70S (PDB code 4V5C) are shown as two vectors corresponding to the two directions of rotation/swivel (blue) and tilt (orange and yellow). (**B**) *Top*, same view as in panel A but showing only the tilt of the head domain. *Bottom*, a 90° rotated view showing the tilt is downward resulting in movement of the head domain away from the body domain. (**C**) *Left*, the same view as in panel A with only the counterclockwise swivel/ rotation of the head domain indicated (left). *Right*, a 90° horizontal rotated view shows that the swivel is greatest toward the subunit interface, close to e*/E tRNA^Pro and on the surface of the ribosome.

compaction of the mRNA means there is no space for the entire four-nucleotide codon in the E site upon full translocation of the mRNA-tRNA pair. Additionally, the mRNA path 5' from the E site turns sharply by ~100° as the mRNA transits to the outside of the ribosome (*Figure 4C*). In the case of e*/ E tRNA^Pro-CGG, upon full translocation of the mRNA-tRNA pair to the E site, the first nucleotide of the codon would be displaced from the E site to accommodate the three-nucleotide codon or the last three nucleotides of the CCC-U codon. This placement would position the mRNA in the +1 frame.

The mRNA boundary between the P and the E sites on the 30S subunit appears to be demarcated by 16S rRNA nucleotide G926 (*Figure 4B and C*; *Selmer et al., 2006*). G926 interacts with the phosphate of the +3 nucleotide of the E-site codon thus defining its 3'-end (*Figure 4C*). In our structure of tRNA^Pro-CGG interacting with a near-cognate, +1 slippery codon, G926 instead interacts with $U_{+4}$ of the CCC-U codon rather than the third nucleotide, thereby defining $U_{+4}$ as the end of the E-site codon (*Figure 4E*). Even in translocation intermediate ribosome structures containing EF-

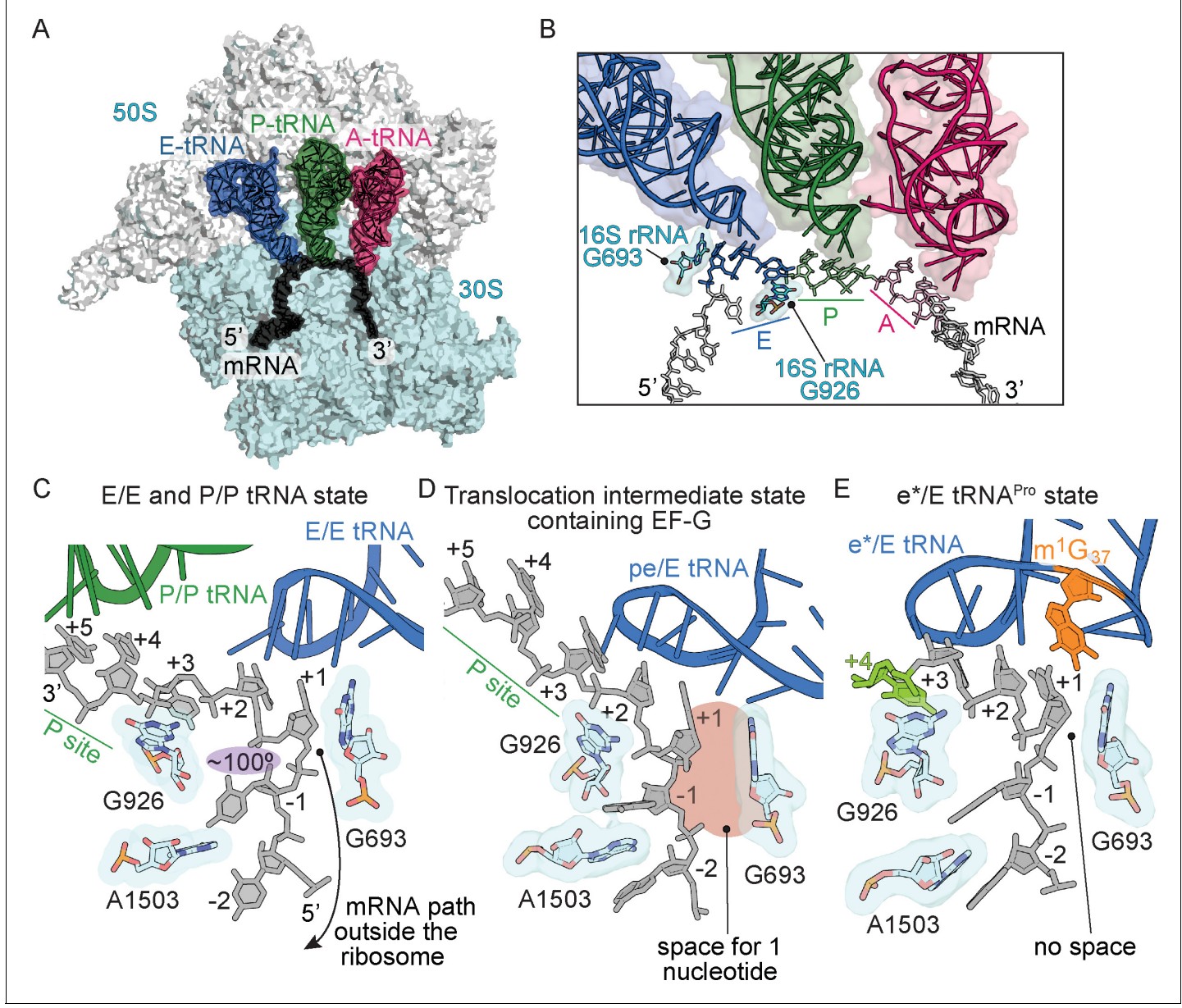

**Figure 4.** Frame-dependent conformations of the mRNA in the E and P sites. (A) Overview of the 70S ribosome with (B) a zoomed-in view of the mRNA-tRNA interaction in the A, P, and E sites (PDB code 4V6F). 16S rRNA nucleotides G693 and G926 interact with the E-site codon–anticodon. (C) The normal path of the mRNA (black) in a ribosome structure containing P/P and E/E tRNAs demonstrates only a three-nucleotide codon (nucleotides +1, +2 and +3) is accommodated in the E site (PDB code 4V5F). 16S rRNA G693 defines the starts of the E-site codon and interacts with the first nucleotide. As the mRNA leaves the E site, there is a 100° kink between the first nucleotide of the E-site codon (+1) and the −1 nucleotide (shaded in purple). Panel C is rotated ~180° relative to the view in panel B. (D) A translocation intermediate structure induced by EF-G contains a tRNA positioned between the P and the E sites on the 30S (denoted 'pe'). The pe/E tRNA has not undergone full translocation to the E site and thus only two nucleotides (+1 and +2) are located in the E site (PDB code 4W29). In this translocation intermediate state, there is space available to accommodate an additional nucleotide of the codon (shaded in red) that would occur upon full translocation. (E) tRNA$^{Pro}$ bound to a +1 slippery CCC-U codon reveals that although the codon–anticodon pair has not been fully translocated, this placement of the mRNA is different as compared to normal translocation intermediates structures as shown in panel D. The additional nucleotide (+4) of the four-nucleotide codon is shown in green.

G, G926 interacts with the +3 phosphate of the mRNA, although this ribosome complex contains a cognate mRNA-tRNA pair (*Figure 4D*; *Zhou et al., 2014*). Comparison of post-translocation (*Gao et al., 2009*), translocation intermediate (*Zhou et al., 2013*; *Zhou et al., 2014*), and our structure presented here reveals that the position of G926 in all three is very similar, while it is the

position of the mRNA that changes substantially (*Figure 4C, D and E*). In summary, although tRNA[Pro]-CGG moves to the E site in our structure presented here, the four-nucleotide mRNA codon is compacted in the E site as if translocation has already taken place (*Figure 4E*).

### The G966-C1400 bridge connecting the 30S head and body domains is broken in the presence of a near-cognate codon–anticodon interaction

The ribosome minimally interacts with the mRNA-tRNA pair in the P site: 16S rRNA C1400 contacts G966 and stacks with the third base-pair of the codon–anticodon (*Figure 5A*; *Selmer et al., 2006*). The G966-C1400 pair remains intact during translocation of the mRNA-tRNA pair from the P to the E site as part of the head domain, as evidenced by ribosome structures in translocation intermediate states containing EF-G (*Figure 5B*). (*Zhou et al., 2013*; *Zhou et al., 2014*). In our structure containing e*/E tRNA[Pro]-CGG bound to a CCC-U codon, the G966-C1400 interaction is ablated (*Figure 5C*). Since the G966-C1400 interaction is effectively the bridge between the head and body domains, this disruption can be attributed to a dysregulation caused by this spontaneous translocation event.

### The absence of the m$^1$G37 modification in tRNA[Pro]-CGG coupled with a +1 slippery codon–anticodon interaction causes disordering of the 30S head domain

Finally, we solved a 4.1 Å structure of tRNA[Pro]-CGG lacking m$^1$G37($\Delta$m$^1$) and interacting with a near-cognate, slippery CCC-U codon at the P site. The electron density for the majority of the tRNA[Pro]-CGG on the 50S subunit is well-resolved, indicating that, similar to the analogous structure

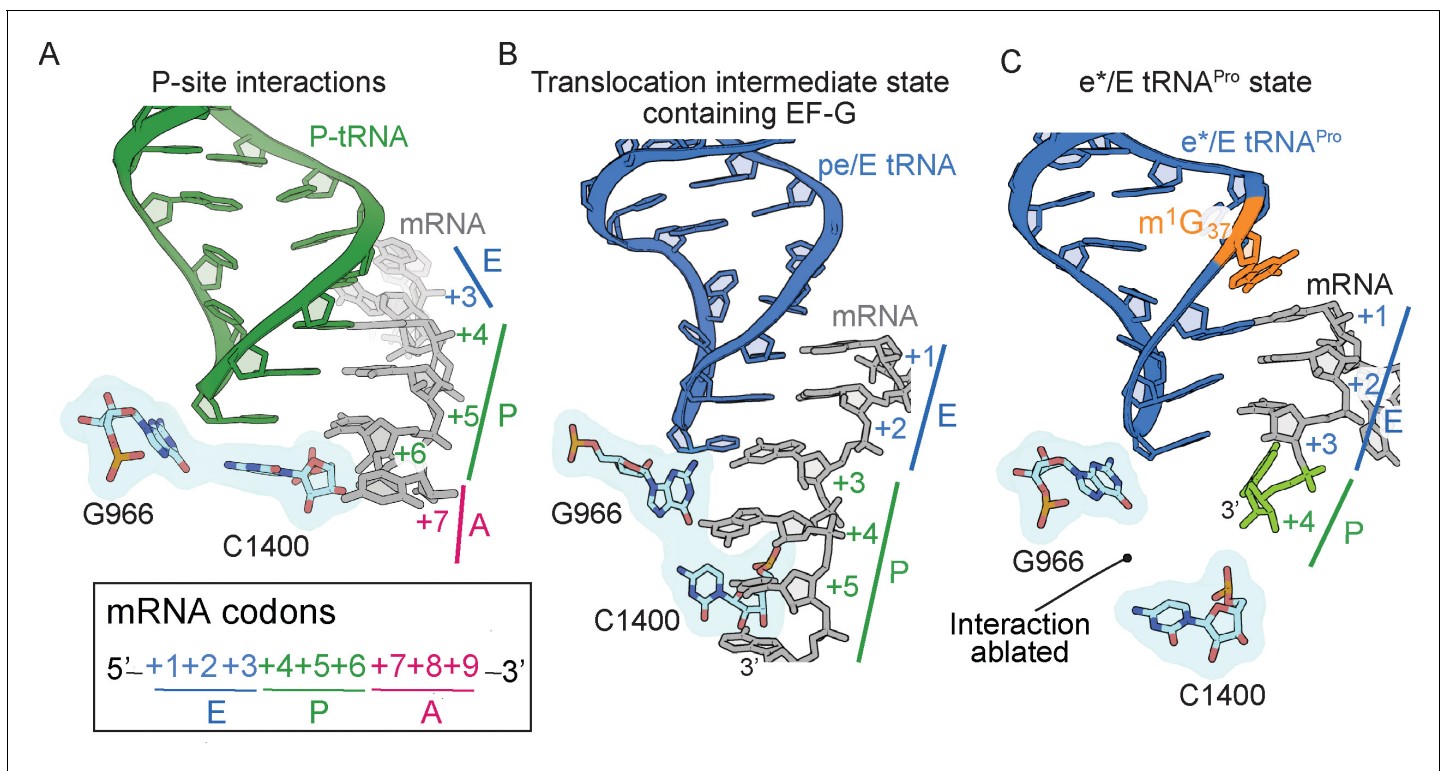

**Figure 5.** The 16S rRNA G966-C1400 gate between the 30S head and body domains is disrupted during a frameshift event. (**A**) In a post-translocation state containing E/E, P/P and A/A tRNAs, 16S rRNA nucleotides G966 and C1400 are located beneath the P-site tRNA (PDB code 4V5F). (**B**) The G966 and C1400 interaction remains intact during EF-G-mediated translocation as the position of 30S head domain nucleotide G966 shifts while 30S body nucleotide C1400 remains constant (PDB code 4W29). (**C**) In a ribosome undergoing a +1 frameshift induced by tRNA[Pro] and a slippery CCC-U codon, the G966-C1400 interaction is broken. The additional nucleotide (+4) of the four-nucleotide codon is shown in green.

The online version of this article includes the following figure supplement(s) for figure 5:

**Figure supplement 1.** 30S interactions with the P-site ASL[Pro].

solved in the presence of the $m^1G37$ modification (*Figure 2C*), the tRNA has also moved towards the E site on the 50S subunit (*Figure 2D*). In contrast, the electron density is weak for the tRNA[Pro]-CGG ASL and E-site mRNA indicating these regions are dynamic in the context of slippery codon and in the absence of $m^1G37$ modification (*Figure 2D*, *Figure 2—figure supplement 2G*). Disordering of tRNA[Pro] $G37(\Delta m^1)$ starts approximately at the beginning of the anticodon loop at nucleotides U32-A38. Correspondingly, the 30S head domain region is also disordered (*Figure 2D*, *Figure 2—figure supplement 3D*), whereas the only region of the 30S body domain with poor electron density is the P/E loop nucleotides G1338 and A1339. Taken together, in this mRNA-tRNA pairing that causes high levels of frameshifting, interactions with the tRNA are destabilized in the P site resulting in the codon–anticodon adopting an e* location and high mobility of the 30S head domain.

## Discussion

The importance of tRNA modifications in protein synthesis has been recognized for decades, yet the precise roles of modifications located outside the anticodon have been elusive. Modifications in the anticodon loop at position 37 of tRNAs have been implicated in mRNA frame maintenance (*Urbonavicius et al., 2001*; *Yarian et al., 2002*), but how the absence of a single methylation can dysregulate the mRNA frame remained unclear. Here, we elucidated the role of $m^1G37$ in tRNA[Pro]-CGG, which was known to be important in stabilizing stacking interactions with anticodon nucleotides during decoding at the A site (*Maehigashi et al., 2014*) and in the prevention of frameshifting (*Björk et al., 1989*; *Hagervall et al., 1993*). We find that this single methyl group influences the overall stability of tRNA[Pro]-CGG on the ribosome in an unexpected manner and causes large conformational changes between the tRNA and the 30S head domain, a domain known to move extensively during translocation of the tRNAs (*Wasserman et al., 2016*; *Ratje et al., 2010*; *Zhou et al., 2013*; *Zhou et al., 2014*). However, the methylation *alone* does not stabilize tRNA[Pro]-CGG on the ribosome and, instead, its position is heavily influenced by interactions with a slippery proline codon. Our structures of different mRNA-tRNA[Pro]-CGG pairs on the ribosome reveals the first detailed mechanistic insight into how mRNA frame maintenance is regulated by both the $m^1G37$ modification and the stability of mRNA-tRNA interaction.

The location on the ribosome where frameshifting can occur is likely dependent on the type of frameshift (in the positive or negative directionon the mRNA) and whether tRNA or mRNA causes the frameshift (*Dinman, 2012*; *Dunkle and Dunham, 2015*; *Atkins et al., 2016*; *Korniy et al., 2019*). The ribosome itself can prevent frameshifts through inherent differences in how the ribosome interacts with the tRNA-mRNA complex at the different tRNA binding sites and such differences could potentially influence frameshifting by relaxing interactions, for example, between the codon and anticodon. While the codon–anticodon interaction located in the 30S A site is strictly monitored by conserved 16S rRNA nucleotides to select for cognate tRNA (*Ogle et al., 2002*), there are comparatively few interactions with the codon–anticodon when positioned within the P and E sites, underscoring their different functional roles during the translation cycle. The relative absence of interactions in the P and E sites provides an opportunity for the mRNA to shift out of frame. Some tRNA[Pro] isoacceptors frameshift either during translocation from the A to the P site or after the translocation step and once positioned in the P site (*Gamper et al., 2015a*; *Gamper et al., 2015b*). Therefore we sought to capture the interactions of tRNA[Pro] during a frameshift event. The structure of ASL[Pro] bound to a slippery codon in the P site reveals the codon–anticodon interaction has shifted into the +1 frame indicating that the anticodon stem-loop interaction with the mRNA codon alone is important for frameshifting (*Figure 1D*). This frameshift is likely possible because interactions with the P-site tRNA are limited to only 16S rRNA P/E loop nucleotides G1338 and A1339 with the anticodon stem and 16S rRNA C1400 with the anticodon nucleotide 34 (*Figure 5A*, *Figure 5—figure supplement 1*). The P/E loop appears to indirectly enforce the mRNA frame by gripping the P-site tRNA and these interactions are maintained as the tRNA moves from the P site to a hybrid P/E state (*Dunkle et al., 2011*) and to a translocation intermediate pe/E state (*Zhou et al., 2013*). In the structures of ASL[Pro] interacting with the mRNA in the 0 frame, there is well-resolved density for G1338 and A1339 (*Figure 1B and C*, *Figure 1—figure supplement 1A,D, G*), while there is some disordering of these nucleotides in the context of ASL[Pro] bound to the near-cognate, slippery codon in the +1 frame (*Figure 1—figure supplement 1C, F and I*). The inability for G1338 and A1339 to grip the anticodon stem when the codon–anticodon is in the +1 frame likely

results in the destabilization of full-length tRNA$^{Pro}$ in these frameshift-competent contexts (*Figure 2C and D*). Simulations of the ribosome undergoing large movements of the head domain during translocation implicate nucleotides G1338 and A1339 in the coupling of 30S head dynamics and displacement of the P-site tRNA towards the E site (*Nguyen and Whitford, 2016*). Therefore, the lack of gripping by the P/E loop in the P site when a destabilized mRNA-tRNA interaction is present, appears to bias tRNA$^{Pro}$ toward the E site which, in turn, causes the 30S head domain to swivel and tilt as if EF-G were bound.

In addition to P/E loop nucleotides G1338 and A1339, 16S rRNA nucleotide G966 is also part of the 30S head domain and stacks with nucleotide 34 of the P-site tRNA anticodon (*Figure 5A*). The 30S head is connected to the body domain via interactions of G966 with nucleotide C1400; the G966-C1400 pair remains stacked beneath the P-site tRNA and follows the tRNA as it moves to a hybrid P/E state and to a translocation pe/E intermediate state upon EF-G binding (*Figure 5B*; *Zhou et al., 2013*; *Zhou et al., 2014*; *Dunkle et al., 2011*). In the case of P-site ASL$^{Pro}$ interacting with a +1 slippery codon in the 0 or +1 frame, *cis*-Watson–Crick interactions form between C$_{+3}$•C34 and U$_{+4}$•C34, respectively (*Figure 1—figure supplement 1K and L*). Both the C$_{+3}$•C34 and U$_{+4}$•C34 interactions appear to result in reduced stacking with C1400. In structures of full-length tRNA$^{Pro}$-mRNA pairs that cause +1 frameshifting and destabilization at the P site, the G966-C1400 interaction is broken (*Figure 5C*). Together, these results suggest a connection between the reduced interactions of G1338-A1339 and G966-C1400 with the tRNA that appear to influence the movement of the P-site tRNA towards the E site.

Although neither C1400 nor G966 have previously been implicated in mRNA frame maintenance, there is a functional link between the P/E loop nucleotides G1338 and A1339 and C1400-G966. 16S rRNA nucleotides in the P site are generally more tolerable of mutations than the A-site 16S rRNA, although substitution of G966 substantially reduces ribosome activity to ~10% despite the mutation not being lethal (*Abdi and Fredrick, 2005*). This G966 mutant is suppressed by a G1338A mutation, indicating that the G1338A mutation can stabilize interactions with the P-site tRNA even in the absence of the C1400-G966 interaction.

The interactions observed between the codon and anticodon of ASL$^{Pro}$ bound to either cognate or near-cognate codons reveal the process of shifting into the +1 frame (*Figure 1D*). In the context of full-length mRNA-tRNA pairs that cause +1 frameshifting, both the tRNA and mRNA move from the P site to occupy a position between the E and the P site (*Figure 2C and D*). This mRNA-tRNA placement is similar to the translocation intermediate containing EF-G (*Zhou et al., 2013*). In the structure presented here containing an e*/E tRNA$^{Pro}$ as compared to the EF-G containing structures containing an pe/tRNA, e*/E tRNA$^{Pro}$ is positioned slightly further away from the mRNA. In the e*/E tRNA$^{Pro}$-CGG structure, the first position of the codon–anticodon forms a Watson–Crick interaction but the second and third nucleotides of the codon–anticodon are not within hydrogen bonding distances (*Figure 2G*). Interestingly, in the EF-G-bound translocation intermediate structures, the codon–anticodon is also not within bonding distance (*Zhou et al., 2013*). These results suggest that it is the disruptive nature of moving between tRNA binding sites that perturbs the interactions between the codon and anticodon and these interactions likely reform once the transition to the next tRNA binding site is complete.

30S head domain conformational changes were first captured in ribosome structures of tRNAs moving between the A to the P sites (ap/ap state) and between the P and the E sites (pe/E state) on the 30S in the presence of EF-G (*Zhou et al., 2013*; *Zhou et al., 2014*; *Dunkle et al., 2011*; *Ratje et al., 2010*). Upon EF-G binding, the head domain is predicted to first tilt and then swivel in a counterclockwise manner (as viewed with the E, P, A tRNA sites from left to right shown in *Figure 2*) to translocate the two tRNAs to final E/E and P/P positions (*Wasserman et al., 2016*; *Nguyen and Whitford, 2016*). The departure of EF-G and reverse tilting of the head back towards the intersubunit space is followed by clockwise swivel, the rate-limiting step for translocation (*Wasserman et al., 2016*). Frameshift-prone tRNAs, such as tRNA$^{Pro}$-CGG, cause spontaneous head swiveling and tilting once the tRNA occupies the P site after the mRNA frameshift has occurred. The inability of the ribosome to hold the P-site tRNA is influenced by weakening of the interactions of the P/E loop with the anticodon stem and by disruption of the C1400-G966 interaction, both events likely major contributors to the dysregulation of the 30S head domain. In other words, the head domain is unable to maintain extensive interactions with the P-site tRNA and this failure leads to spontaneous translocation. Single molecule FRET (smFRET) studies of translocation events also show that upon tRNA

movement from the P to the E site, there is increased dissociation of the tRNA from the ribosome, bypassing the post-translocation E/E state (*Wasserman et al., 2016*). These data seem to suggest that even during canonical translocation from the P to the E site, this is a highly dynamic process which leads to destabilized tRNA. Consistent with these observations, we would expect that once EF-G fully translocates tRNA$^{Pro}$-CGG to the post E/E state, there may be a further decrease in interactions between the tRNA and ribosome. Additionally, our structures show 30S head swiveling and tilting in the absence of EF-G, concurrent with the +1 frameshift event. This 30S head swivel and tilt movement would likely prevent EF-G from binding until the the head resets to a non-rotated state. EF-G residues located in the tip of domain IV interact with the anticodon stem-loop of the A-site tRNA during translocation to the P site (*Zhou et al., 2013*) and mutations in this domain slow the rate of translocation (*Rodnina et al., 1997*; *Savelsbergh et al., 2000*). Recent studies using slippery codon sequences where spontaneous frameshifting occurs, EF-G can restrict frameshifting and helps to maintain the mRNA frame (*Peng et al., 2019*). It is therefore an enticing hypothesis that EF-G may have a role in mRNA frame maintenance in addition to its established function in translocation.

Although all three prokaryotic tRNA$^{Pro}$ isoacceptors have the capacity to frameshift, whether they use similar mechanisms of action is unknown (*Björk et al., 1989*; *Gamper et al., 2015a*; *Qian et al., 1998*; *O'Connor, 2002*). The m$^1$G37 modification minimally influences frameshifting in proline isoacceptor tRNA$^{Pro}$-GGG (*proL*) in contrast to tRNA$^{Pro}$-cmo$^5$UGG (*proM*) (*Gamper et al., 2015a*). An additional difference is the dependency on elongation factor-P (EF-P) to reconcile +1 frameshifts. EF-P binds at the E site to overcome ribosome stalling induced by poly-proline codons (*Ude et al., 2013*; *Huter et al., 2017*) and is critical in suppressing frameshifts on poly-proline stretches (*Gamper et al., 2015a*). While EF-P reduces +1 frameshifts with tRNA$^{Pro}$-GGG G37($\Delta$m$^1$) to an equivalent frequency as native tRNA$^{Pro}$-GGG, the absence or presence of m$^1$G37 has little influence on the ability of isoacceptor tRNA$^{Pro}$-cmo$^5$UGG to frameshift (*Gamper et al., 2015a*). These mechanistic differences may be due to a combination of the codon–anticodon pairings on slippery codons (tRNA$^{Pro}$-cmo$^5$UGG and tRNA$^{Pro}$-GGG are cognate with slippery codons while tRNA$^{Pro}$-CGG is near-cognate [*Nasvall et al., 2004*]) and/or the influence of other modifications such as the cmo$^5$U34 modification in tRNA$^{Pro}$-cmo$^5$UGG (*Masuda et al., 2018*). Further considerations may be the location of the slippery codon on the mRNA, which would have varying nascent chain lengths, and whether the following codon after the slippery codon is rare (*Gamper et al., 2015a*). Our structures show that both the m$^1$G37 modification status of tRNA$^{Pro}$-CGG and the CCC-U codon causes the tRNA to become destabilized and its position is biased towards the E site, which we predict is indicative of high levels of frameshifting. The possible synergistic effects of cmo$^5$U34 and m$^1$G37 in tRNA$^{Pro}$-cmo$^5$UGG in preventing frameshifts is unclear, but tRNA$^{Pro}$-UGG lacking all modifications exhibits higher levels of frameshifting as compared to tRNA$^{Pro}$-UGG lacking only the m$^1$G37 modification (*Gamper et al., 2015b*). Since the presence of m$^1$G37 in tRNA$^{Pro}$-cmo$^5$UGG may restrict its movement to the e*/E position, this tRNA isoaccepor may no longer be an optimal substrate for EF-P, which is consistent with kinetic analyses (*Gamper et al., 2015a*). Together, our structures of tRNA$^{Pro}$-CGG in frameshifting contexts provides new insights into how RNA modifications impact tRNA stability on the ribosome.

## Materials and methods

### mRNA, ASL, and ribosome purification

ASL$^{Pro}$ containing a m$^1$G37 modified anticodon stem-loop (17 nucleotides) and mRNA (19 nucleotides) were purchased from GE Healthcare Dharmacon and dissolved in 10 mM Tris-HCl, pH 7.0, and 5 mM MgCl$_2$. We used a chemically synthesized ASL to ensure complete m$^1$G modification at position 37, as previously used to examine interactions in the A site (*Maehigashi et al., 2014*). Purification of *Thermus thermophilus* 70S ribosomes was performed as previously described (*Zhang et al., 2018*).

### tRNA$^{Pro}$-CGG ligation and purification

To ensure that tRNA$^{Pro}$-CGG was methylated at tRNA nucleotide 37 (m$^1$G), the 5' half of the tRNA (nucleotides 1–39) was chemically synthesized (GE Healthcare Dharmacon) and enzymatically ligated to the chemically synthesized 3' half following established protocols (*Sherlin et al., 2001*; *Stark and*

*Rader, 2014*). Briefly, T4 polynucleotide kinase (NEB) was used to phosphorylate the 5' end of the 3' tRNA half and was heat inactivated. The 5' and 3' halves of each tRNA were then mixed and annealed in the T4 RNA ligase buffer by heating to 80°C for five min and slow cooling on the heat block to room temperature. T4 RNA ligase (NEB) was added to the reaction at 37°C for 18 hr. The ligation reaction was run on a 12% denaturing 8M urea-polyacrylamide gel and the ligated fragment was excised and purified using a modified crush and soak method (*Stark and Rader, 2014*). The RNA was ethanol precipitated, the pellet was thoroughly air dried and resuspended in 10 mM Tris-HCl, pH 7.0 and 5 mM $MgCl_2$, followed by annealing at 70°C for 2 min and slow cooled to room temperature on the benchtop. The purified full-length RNA was aliquoted and stored at $-20$°C.

## Structural studies

$ASL^{Pro}$ complexes were formed with 3.5 μM 70S ribosomes programmed with 7 μM mRNA for 6 min at 37°C. Then 22 μM $ASL^{Pro}$ was added and incubated for 30 min at 55°C. $tRNA^{Pro}$-CGG and $tRNA^{Pro}$-CGG G37 ($\Delta m^1 G$) complexes were formed with 3.5–4.4 μM of 70S ribosomes programmed with 7–10.5 μM mRNA for 6 min at 37°C. Then 7.7–10.5 μM $tRNA^{Pro}$-CGG or $tRNA^{Pro}$-CGG G37 ($\Delta m^1 G$) were incubated for 30 min at 55°C. Each ASL and tRNA were positioned in the ribosomal P site by designing the Shine-Dalgarno sequence eleven nucleotides upstream of the P-site codon. Crystals were grown by sitting-drop vapor diffusion in a 1:1 drop ratio of 100 mM Tris–acetate pH 7.6, 12–13% 2-methyl-2,4-pentanediol (MPD), 2.9–3.0% polyethylene glycol (PEG) 20K, 100–150 mM L-arginine-HCl and 0.5–1 mM β-mercaptoethanol (β-Me). $ASL^{Pro}$ crystals were cryoprotected stepwise in 100 mM Tris–acetate pH 7.6, 10 mM $NH_4Cl$, 50 mM KCl, 3.1% PEG 20K, 10 mM Mg $(CH_3COO)_2$, 6 mM β-Me and 20–40% MPD, with the final cryoprotection solution containing 30 μM $ASL^{Pro}$. Cryoprotection of $tRNA^{Pro}$-CGG and $tRNA^{Pro}$-CGG G37 ($\Delta m^1 G$) was accomplished similarly to $ASL^{Pro}$, except with 3% PEG 20K and 20 mM $Mg(CH_3COO)_2$. Additionally, the final cryoprotection solution for both $tRNA^{Pro}$-CGG and $tRNA^{Pro}$-CGG G37 ($\Delta m^1 G$) did not contain ligand. All crystals were flash frozen in liquid nitrogen for data collection.

X-ray diffraction data were collected at the Southeast Regional Collaborative Access Team (SER-CAT) 22-ID beamline line and the Northeastern Collaborative Access Team (NE-CAT) ID24-C and ID24-E beamlines at the Advanced Photon Source (APS). Data were integrated and scaled using the program XDS (*Kabsch, 2010*). All structures of 70S-$ASL^{Pro}$- and $tRNA^{Pro}$-CGG bound to cognate mRNA were solved by molecular replacement in PHENIX using coordinates from a 70S structure containing mRNA and tRNAs (PDB code 4V6G) (*Jenner et al., 2010*; *Adams et al., 2010*). For the 70S complex containing an e*/E tRNA and mRNA, the start model was changed to $tRNA^{SufA6}$ bound to a slippery sequence (PDB code 5VPP) (*Hong et al., 2018*). In this structure, the 30S head domain is swiveled ~18°. The structure was solved by molecular replacement in PHENIX followed by iterative rounds of manual building in Coot (*Emsley et al., 2010*). All figures were prepared in PyMOL (*Schrodinger LLC, 2010*).

## Calculating 30S head movement

The following protocol was used to describe the 30S head and body domains in terms of Euler angles (φ,ψ,θ). Here, φ+ψ defines the net swivel angle, tilting is described by θ and the tilt direction is defined by φ. Euler angles were calculated separately for the 16S body and head domains as previously defined (*Nguyen and Whitford, 2016*). Briefly, for all calculations, the P atoms of the 'core' residues of the 23S rRNA and 16S head/body were considered. The core residues are defined as the set of atoms that were found to have spatial root mean squared fluctuations of less than 1 Å in a one microsecond explicit-solvent simulation (*Whitford et al., 2013*). This includes 1351, 442, and 178 residues in the 23S rRNA, 16S body, and 16S head, respectively. To ensure that the calculated angles reflect global domain orientations, rather than local deformations, reference configurations of the 23S, as well as the 16S head and body, were aligned to each group of core residues, separately. To calculate 16S body swivel angles, each model was first aligned to a crystallographic model of a classical unrotated ribosome (PDB codes 4V9D:DA and 4V9D:BA), where alignment was based on the 23S rRNA core atoms. To define the orientation of the body, axes were constructed based on the aligned orientations of residues 41, 127, and 911, which are used to define the plane of rotation. The Euler angles are then calculated based on the orientational difference of the fitted axes in the reference classical configuration and the structural model of interest. To describe the orientation of

the head, the 16S body was first aligned to the reference model (alignment based on core atoms) and axes were defined based on residue 984, 940 and 1106. According to these definitions, models (4V9D:DA, 4V9D:BA) and (4V9D:CA, 4V9D:AA) define pure (tilt-free) body rotation, and models (4V9D:DA, 4V9D:BA) and (4V4Q:DB, 4V4Q:CA) define pure (tilt-free) head swivel. The classical model (4V9D:DA, 4V9D:BA) is defined as the zero-tilt, zero-rotation orientation.

## Acknowledgements

We thank Ha An Nguyen, Ian Pavelich, Jacob Mattingly, Pooja Srinivas, Dr. Alexandra Kuzmishin Nagy and Dr. Graeme L Conn for critical reading of the manuscript, and the staff members of the NE-CAT and SER-CAT beamlines for assistance during data collection. This work was supported by the NIH R01GM093278 (to CMD), NIH R01GM119386 (to RLG Jr), and NSF MCB-1915843 (PCW). CMD is a Burroughs Wellcome Fund Investigator in the Pathogenesis of Infectious Diseases. The X-ray crystallography datasets were collected at the Northeastern Collaborative Access Team (NE-CAT) and Southeast Regional Collaborative Access Team (SER-CAT) beamlines. At the NE-CAT beamlines (GM124165), a Pilatus detector (RR029205), and an Eiger detector (OD021527) were used. At SER-CAT, the beamlines is supported by its member institutions, and equipment grants (S10RR25528, S10RR028976 and S10OD027000) from the NIH. This research used resources of the Advanced Photon Source, a U.S. Department of Energy (DOE) Office of Science User Facility operated for the DOE Office of Science by Argonne National Laboratory under Contract No. DE-AC02-06CH11357 (NE-CAT) and Contract No. W-31-109-Eng-38 (SER-CAT).

## Additional information

### Funding

| Funder | Grant reference number | Author |
| --- | --- | --- |
| National Institutes of Health | R01GM093278 | Christine M Dunham |
| National Institutes of Health | R01GM119386 | Ruben L Gonzalez Jr |
| National Science Foundation | MCB-1915843 | Paul C Whitford |

The funders had no role in study design, data collection and interpretation, or the decision to submit the work for publication.

### Author contributions

Eric D Hoffer, Formal analysis, Validation, Investigation, Visualization; Samuel Hong, Conceptualization, Formal analysis, Validation, Visualization, Methodology; S Sunita, Conceptualization, Data curation, Formal analysis, Supervision, Investigation; Tatsuya Maehigashi, Conceptualization, Data curation, Formal analysis; Ruben L Gonzalez Jnr, Writing - review and editing; Paul C Whitford, Funding acquisition, Investigation, Methodology; Christine M Dunham, Conceptualization, Formal analysis, Supervision, Funding acquisition, Validation, Investigation, Visualization, Methodology

### Author ORCIDs

Ruben L Gonzalez Jnr  https://orcid.org/0000-0002-1344-5581
Christine M Dunham  https://orcid.org/0000-0002-8821-688X

### Decision letter and Author response

Decision letter https://doi.org/10.7554/eLife.51898.sa1
Author response https://doi.org/10.7554/eLife.51898.sa2

## Additional files

### Supplementary files

• Transparent reporting form

## Data availability

Crystallography, atomic coordinates, and structure factors have been deposited in the Protein Data Bank, www.pdb.org (PDB codes 6NTA, 6NSH, 6NUO, 6NWY, 6O3M, 6OSI).

The following datasets were generated:

| Author(s) | Year | Dataset title | Dataset URL | Database and Identifier |
|---|---|---|---|---|
| Hoffer ED, Hong S, Sunita S, Maehigashi T, Dunham CM | 2020 | Modified ASL proline bound to Thermus thermophilus 70S (cognate) | https://www.rcsb.org/structure/6NTA | RCSB Protein Data Bank, 6NTA |
| Hoffer ED, Hong S, Sunita S, Maehigashi T, Dunham CM | 2020 | Modified ASL proline bound to Thermus thermophilus 70S (near-cognate, +1 sliipery codon) | https://www.rcsb.org/structure/6NSH | RCSB Protein Data Bank, 6NSH |
| Hoffer ED, Hong S, Sunita S, Maehigashi T, Dunham CM | 2020 | Modified tRNA(Pro) bound to Thermus thermophilus 70S (cognate) | https://www.rcsb.org/structure/6NUO | RCSB Protein Data Bank, 6NUO |
| Hoffer ED, Hong S, Sunita S, Maehigashi T, Dunham CM | 2020 | Modified tRNA(Pro) bound to Thermus thermophilus 70S (near-cognate, +1 slippery codon) | https://www.rcsb.org/structure/6NWY | RCSB Protein Data Bank, 6NWY |
| Hoffer ED, Hong S, Sunita S, Maehigashi T, Dunham CM | 2020 | Unmodified tRNA(Pro) bound to Thermus thermophilus 70S (cognate) | https://www.rcsb.org/structure/6O3M | RCSB Protein Data Bank, 6O3M |
| Hoffer ED, Hong S, Sunita S, Maehigashi T, Dunham CM | 2020 | Unmodified tRNA(Pro) bound to Thermus thermophilus 70S (near cognate, +1 slippery codon) | https://www.rcsb.org/structure/6OSI | RCSB Protein Data Bank, 6OSI |

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
