## [Decision Letter]

**Acceptance summary:**

This paper by Dunham and coworkers uses structural biology to determine how post-transcriptional methylation of a tRNA nucleotide (m^1^G37) helps maintain the mRNA reading frame and suppress frameshifting. They determine six crystal structures of the bacterial ribosome engaged with tRNA in the absence or presence of the m^1^G37 tRNA modification and bound to either cognate or slippery codons. Comparison of the structures reveals that the m^1^G37 methyl group stabilizes the tRNA on the ribosome in the presence of a slippery sequence, providing new mechanistic insights into how mRNA frame maintenance is regulated by both tRNA modifications and mRNA:tRNA interactions.

**Decision letter after peer review:**

[Editors’ note: the authors submitted for reconsideration following the decision after peer review. What follows is the decision letter after the first round of review.]

Thank you for submitting your work entitled "Structural basis of mRNA +1 frameshifting by tRNAPro CGG on a near-cognate, slippery codon" for consideration by *eLife*. Your article has been reviewed by three peer reviewers, and the evaluation has been overseen by a Reviewing editor and the Senior Editor. The reviewers have opted to remain anonymous.

Our decision has been reached after consultation between the reviewers. Based on these discussions and the individual reviews below, we regret to inform you that your work will not be considered further for publication in *eLife*.

The reviewers were impressed with the range of structures presented but had concerns about the physiological relevance of these structures, which they felt could not be fully addressed in two months without substantial supporting evidence. They also felt the paper would require a substantial rewrite to make it accessible to readers.

Reviewer #1:

Previous work from the Dunham group showed that the absence of the tRNA m^1^G37 modification (a known suppressor of 1+ frameshifting) destabilizes binding of tRNA(Pro) to the ribosomal A site and results in the ribosome being unable to distinguish between cognate CCG and near-cognate CCC-U codons (PNAS, 2014: 111, 12740-12745)(J. Biol. Chem. 2019, 294:5281-5291). However, the shift into the +1 frame does not occur in the A site. Here, the authors attempt to address the mechanism of frameshifting by determining six structures in which either anticodon stem-loops (ASLs) or full-length tRNAs are programmed into the ribosomal P site. The crystallography is well done with resolutions between 3.1 and 4.1 Å and statistics consistent with other ribosomal structures. The structures show that in the presence of the near-cognate codon, tRNA(Pro) is destabilized and moves towards the ribosomal E site, causing movement of the 30S head domain reminiscent of the movement that occurs during EF-G-mediated translocation.

1) The crystallographic approach potentially suffers from a lack of physiological relevance as it remains unclear how well a complex containing deacylated tRNA in the P site, no polypeptide in the exit tunnel, no A-site tRNA, and an exceptionally short mRNA in the mRNA channel represents the situation of frameshifting in vivo. It is possible that the additional constraints imposed by nascent polypeptide chains and mRNA sequences could change the dynamics and therefore the behavior of the tRNA:mRNA complex. The physiological relevance of these complexes should be addressed.

2) The authors show that head destabilization is more pronounced in the absence of the tRNA methylation, but the exact mechanism for this additional instability was not clear to me. The authors need to address the relevance of the m^1^G37 modification to the suppression of frameshifting and link the structures better to the prior work showing the frequencies of frameshifting in the presence and absence of the m^1^G37 modification.

3) Are the codon:anticodon interactions similar in the ASL and full-length tRNA containing structures? It was difficult to understand if this was true comparing Figures 1 and S4. A figure comparing their positions should be shown. If the interactions are different, does this diminish the relevance of the ASL-containing structures?

4) The schematic model presented in Figure 5 is difficult to decipher. Steps are either missing or unnecessarily complicated. For example, why does the color of the 30S head domain change?

Reviewer #2:

The paper presents several crystal structures of tRNA^Pro^ (ASL or full-length) on its cognate codon and a near-cognate codon in a slippery context that induces +1 frameshifting. The authors compare tRNAs with and without the modification m^1^G37 to understand the effect of the modification on frameshifting.

1) The main solid conclusion of the paper is that even the modified tRNA^Pro^ can frameshift on a near-cognate/slippery codon. This is nice, but the conclusions come from the experiments with the ASLs. The question I have is whether this agrees with the previous functional analysis, i.e. what is the frameshifting frequency of tRNA^Pro^ on CCU/C in cells? Are the ASL structures biologically relevant?

2) The authors present structures with ASLs and the full-length tRNAs. I am not sure I understand why show ASLs if similar information is available from the tRNAs. this has to be clearly described.

3) I am not sure how relevant are the structures with a single deacylated tRNA in the P site or in the e*/E state. The authors indicate that this state represents ribosomes after peptide release. This is correct, but irrelevant for frameshifting. The rotated swiveled state of the ribosome with the +1 frameshifting tRNA is very interesting, but what is the evidence that it occurs in cells?

4) The model in Figure 5 is entirely speculative and misleading with respect to the fate of the E-site tRNA. In state 1 the E-site tRNA is indicated which then dissociates upon action of EF-G. Looking at the recent papers of Puglisi, Rodnina and Cooperman, the favored scenario, at least during normal translation, is that E-site tRNA dissociates before or simultaneously with the ternary complex binding. This should be changed in the model. Then, the +1 frameshifting pathway is entirely speculative. The structures in this paper do not have a tRNA in the E site – so why do the authors include it? It is also not clear from the existing data at which step of translocation +1 frameshifting takes place and to the best of my knowledge there is no independent evidence in support the model.

5) The conclusion as to the importance of the m^1^G37 modification is not clearly discussed.

6) The paper is not well written. It is full of imprecise statements such as "programmed by tRNA" (the ribosome can be programmed by mRNA, but not by tRNA) or 5' anticodon stem (I don't know what this means). The Results include large parts of text than definitely belong to the Introduction, such as the beginning of section 1 and 3. Parts of the Results are speculations that may be more appropriate for the Discussion. The text describing Figures 4 and 5 is very technical and very long and the conclusions are unclear. Numbering of the supplemental figures does not correspond to the numbering in the text. Some of the supplemental figures would be better presented as main figures, e.g. Figures 1 and 2 are not very informative, could be strengthened by combining with the respective supplementary figures. The Abstract is difficult to understand due to the lack of precision.

Reviewer #3:

In this rich but dense paper, the authors discuss six crystal structures of ribosomes with ASL (2: cognate and slippery codons) or tRNA^Pro^ (4: cognate and slippery codons with and without modification at G37) programmed at the P site in order to study mechanistically frameshifting. This is remarkable. They show that the slippery codon is more critical for frameshifting than m^1^G37 modification. This kind of article is extremely difficult to peer-review in the absence of the availability of, for example, Pymol sessions where one can visualize the whole regions of interest easily. We can only rely on the static figures selected by the authors (and without stereo views). I would have liked to have the equivalent of Figure 1 for the six structures. This would have helped during the comparisons. In Figure 1D, U32 is positioned next to a G base, is this right? The size is also not the same as in B and C. It would be nice to see better what C+1 is doing; no contact with G37? Since the authors used chemically synthetized ASL, it is surprising they did not look at crystal structures of ASL without G37 modification (no crystals maybe?). In Supplemental Figure 3, why is the drawing for near cognate in the absence of G37 modification not shown (disorder?).

[Editors’ note: further revisions were suggested prior to acceptance, as described below.]

Thank you for resubmitting your work entitled "Structural insights into mRNA reading frame regulation by tRNA modification and slippery codon-anticodon pairing" for further consideration by *eLife*. Your revised article has been evaluated by Cynthia Wolberger (Senior Editor) and a Reviewing Editor.

The manuscript has been improved but there are some remaining issues that need to be addressed before acceptance, as outlined below:

Revisions:

1) The physiological relevance of the findings, given the use of deacylated tRNA in the P site and a very short mRNA used for the experiments in the paper, is still not clear.

The authors argue that +1 frameshifting is promoted by the tRNA itself, which justifies the use of the minimal model system, and state that the presence of an A-site tRNA and the nascent peptide do not matter. The authors cite Phelps et al., 2006 and Walker and Fredrick, 2006 as support for this statement. However, these papers, and the previous work of the authors, are in vitro studies carried out with deacylated tRNAs and there is no systematic analysis of whether or not the A-site tRNA and nascent peptide affect the frameshifting efficiency. The work by Björk, Wikstrom and Bystrom, 1989, has been carried out in the background of the SufA6 suppressor, rather than with a native tRNA(Pro). In contrast, the paper by Gamper et al., 2015, does not support the statements of the authors. Gamper et al. show that significant +1 frameshifting on a CCCC sequence (which is likely more "slippery" than the CCCU sequence used here) not only depends on the absence of the m^1^G37 modification, but also on the absence of EF-P and the position of the slippery sequence in the gene. Even in the absence of the m^1^G37 modification, the frameshifting efficiency is very low (<2%) in the presence of EF-P and in the middle of a gene (i.e. at physiological conditions), and is only stimulated at particular conditions, e.g. when the slippery codon is engineered at position 2 of the gene or is followed by a rare codon (Figure 1 in the Gamper et al. paper). Thus, although the CCC C/U sequence can induce frameshifting in the absence of the m^1^G37 modification, the efficiency obviously depends on the nascent peptide (middle gene positions vs. position 2) and the presence of the A-site tRNA (demonstrated by the rare codon effects in the A site). The authors should clearly acknowledge these previous findings. The issue with a very short mRNA is even more serious, because a short mRNA may be much more dynamic than that of a longer construct and adopt conformations that are not populated by a longer mRNA.

2) Revision to the Abstract are needed. First sentence should probably read "Modifications in the tRNA adjacent to the three-nucleotide anticodon". The sentence "by stabilizing the tRNA to allow for accurate reading…" is misplaced, because this is a result of the work, which is indicated in the fourth sentence (…destabilizes interactions…). Replace "peptidyl site" with the "P-site of the ribosome". Remove or rephrase the part of the final sentence, as it is not reflecting the results of the work in an adequate way (the slippery codons are known to affect frameshifting).

3) Although the CCCU codon can promote frameshifting in some contexts it is not slippery by itself. The recommendation is to better explain this in the Introduction and use "CCCU codon" throughout the text, rather than "near-cognate, slippery codon".

---

## [Author Response]

[Editors’ note: The authors appealed the original decision. What follows is the authors’ response to the first round of review.]

Reviewer #1:Previous work from the Dunham group showed that the absence of the tRNA m^1^G37 modification (a known suppressor of 1+ frameshifting) destabilizes binding of tRNA(Pro) to the ribosomal A site and results in the ribosome being unable to distinguish between cognate CCG and near-cognate CCC-U codons (PNAS, 2014: 111, 12740-12745)(J. Biol. Chem. 2019, 294:5281-5291). However, the shift into the +1 frame does not occur in the A site. Here, the authors attempt to address the mechanism of frameshifting by determining six structures in which either anticodon stem-loops (ASLs) or full-length tRNAs are programmed into the ribosomal P site. The crystallography is well done with resolutions between 3.1 and 4.1 Å and statistics consistent with other ribosomal structures. The structures show that in the presence of the near-cognate codon, tRNA(Pro) is destabilized and moves towards the ribosomal E site, causing movement of the 30S head domain reminiscent of the movement that occurs during EF-G-mediated translocation.1) The crystallographic approach potentially suffers from a lack of physiological relevance as it remains unclear how well a complex containing deacylated tRNA in the P site, no polypeptide in the exit tunnel, no A-site tRNA, and an exceptionally short mRNA in the mRNA channel represents the situation of frameshifting in vivo. It is possible that the additional constraints imposed by nascent polypeptide chains and mRNA sequences could change the dynamics and therefore the behavior of the tRNA:mRNA complex. The physiological relevance of these complexes should be addressed.

We agree that solving a physiologically-relevant structure is always the goal in the structures we solve. In this study, we focus on specific tRNAs that alone cause frameshifting. There is prior published biochemical evidence that the tRNA itself and, more specifically, the anticodon stem-loop (ASL) *alone*, has the ability to undergo +1 frameshifting (Phelps et al, 2006 and Walker and Frederick, 2006; Hong et al., 2018, is a manuscript from our lab of structures of a frameshift suppressor tRNA^SufA6^, a derivative of tRNA^Pro^, bound to the P site). In the first submission, we described these previous studies in the first sentence of the Results and Discussion section where we explain that this is the rationale for pursuing these structural studies (and we reference prior studies 33 and 34 (which are Phelps et al., 2006 and Walker and Fredrick, 2006)). In these previous studies, both deacylated tRNAs and ASLs were used demonstrating that a P-site tRNA in the absence of a nascent chain, the absence of an A-site tRNA, and the mRNA length all have no meaningful influence on +1 frameshifting in these particular cases. Thus, the only thing required for frameshifting is the ASL of the tRNA itself. These data are the biochemical basis for our structures. We have now introduced this important concept earlier in the manuscript to ensure there is no ambiguity regarding what causes +1 frameshifting by such tRNAs as shown below:

“In circumstances where mRNA frameshifting is caused by changes in the tRNA such as the absence of modifications or changes in the size of the anticodon loop as found in frameshift suppressor tRNAs, primer extension assays demonstrated that the shift into the +1 frame is observed upon direct tRNA binding at the P site (Phelps et al., 2006, Walker and Frederick, 2006). These studies clearly demonstrate that the nature of the interactions between the frameshift-prone mRNA-tRNA pair and the ribosomal P site directly permit frameshifting. Furthermore, the presence of a nascent polypeptide chain, the acylation status of the tRNA (deacylated or aminoacylated), and even the presence of a tRNA in the A site do not contribute to the ability of the tRNA to cause frameshifting.”

More generally, there are at least 50 crystal structures of ribosomes containing a deacylated tRNA at the P site that show normal positioning and engagement with the P site of the ribosome (meaning a P/P tRNA; Noller, Yusupov, Ramakrishnan, Cate, Korostelev, and Polikanov labs and our lab). Therefore, there is clearly strong precedence for ribosome structures containing deacylated tRNAs and adopting P/P positions. tRNA^Pro^ containing m^1^G37 and bound to a cognate CCG codon adopts this canonical position (Figure 2A). This is our control structure. However, when tRNA^Pro^ interacts with a near-cognate codon OR a near-cognate codon and the tRNA^Pro^ lacks the m^1^G37 modification, then we observe destabilization of the tRNA with the P site. Finally, even if we were to aminoacylate P-site tRNA^Pro^, the aminoacyl attachment to the tRNA is extremely labile and would be rapidly deacylated during the lifetime of our experiments, a fact commonly known in the field.

2) The authors show that head destabilization is more pronounced in the absence of the tRNA methylation, but the exact mechanism for this additional instability was not clear to me. The authors need to address the relevance of the m^1^G37 modification to the suppression of frameshifting and link the structures better to the prior work showing the frequencies of frameshifting in the presence and absence of the m^1^G37 modification.

Prior data has clearly shown that the m^1^G37 methylation is needed for mRNA frame maintenance (Gamper et al., 2015). We have extensively updated our Introduction to make sure this point is clear as shown below:

“The absence of m^1^G37 in tRNA^Pro^ causes high levels of +1 frameshifting on so-called “slippery” or polynucleotide mRNA sequences where four nucleotides encode for a single proline codon (Bjork, Wikstrom and Bystrom, 1989). In the case of the tRNAPro CGG isoacceptor, a +1 slippery codon consists of the CCC proline codon and either an additional C or U to form a four-nucleotide codon, CCC-(U/C) (Sroga et al., 1992; Qian et al., 1998) (codons depicted 5’- to 3’). tRNAPro CGG lacking m^1^G37 results in the ribosome being unable to distinguish a correct from an incorrect codon-anticodon interaction during decoding at the aminoacyl (A) site (Nguyen, Hoffer and Dunham, 2019; Maehigashi et al., 2014).”

However in the two structures (out of the six we solved in this paper), we show that the 30S head domain undergoes swiveling and tilting in specific ribosome contexts that are known to cause mRNA frameshifting. Our structures demonstrate it is not only the m^1^G37 methylation that is important, but also the near-cognate, slippery codon-anticodon interaction.

For example, when the interaction between the codon and anticodon is near-cognate and tRNA^Pro^ still contains the m^1^G37 modification, destabilization of tRNA^Pro^ occurs, causing the tRNA to not be gripped well by the ribosomal P site. The tRNA then moves towards the E site. This movement, in turn, causes the 30S head domain to swivel/tilt. In this structure, we are able to discern all features of the model in the electron density. However, when tRNA^Pro^ lacks the m^1^G37 modification and engages in a near-cognate codon-anticodon interaction, we know the tRNA has fully moved to a similar position, but the anticodon is not resolvable due to likely being mobile. Likewise, we know the 30S head domain has moved because we can see the locations of other features of the 30S, but we can’t build the model because, again, the density is poor likely because this part of the 30S is mobile. This emphasized to us the need to solve multiple, different structure to reconcile these differences, which we have done. We have completely rewrote the Discussion to emphasize these results as shown below:

“…We find that this single methyl group influences the overall stability of tRNAPro CGG in an unexpected manner that causes large conformational changes between the tRNA and the 30S head domain, a domain known to move extensively during translocation of the tRNAs (Wasserman et al., 2016; Ratje et al., 2010; Zhou et al., 2013, 2014). However, the methylation *alone* does not stabilize tRNAPro CGG on the ribosome and, instead, its position is heavily influenced by interactions with a distinct slippery proline codon. Our structures of different mRNA-tRNAPro CGG pairs on the ribosome allows the first detailed mechanistic insight into how mRNA frame maintenance is regulated by both the m^1^G37 modification and the stability of mRNA-tRNA interaction.”

3) Are the codon:anticodon interactions similar in the ASL and full-length tRNA containing structures? It was difficult to understand if this was true comparing Figures 1 and S4. A figure comparing their positions should be shown.

We have now combined Figures 2 and S4 (these are the same structures; the reviewer suggested combining structures in Figures 1 and S4 but these are different structures). See the updated Figure 2.

The 70S-ASL structures in Figure 1 show that the ASL is located in the P site while the 70S-tRNA structures show that, in specific frameshifting contexts, the tRNA has moved from the P to almost the E site (i.e. e*/E site). However, it must be noted that to compare the two 70S-ASL structures with the four 70-tRNA structures is difficult due to the nature of the swivel and tilt of the 30S head domain. Briefly, as the tRNAs are translocating from the A and P to the P and E sites, the interactions between the codon and the anticodon are broken even when this interaction is cognate as seen in structures by the Noller group (Shou et al., 2019). So while we do see a slight disruption in the codon-anticodon interaction of the 70S-e*/E tRNA^Pro^-CCC-U codon, this isn’t really the major point. The more important points are the changes to the mRNA path which is what we emphasize in the current manuscript and in Figures 4 and 5. We have now expanded and compared the interactions of the codon-anticodons between these structures.

“The interactions observed between the codon and anticodon of ASL^Pro^ bound to either cognate or near-cognate codons reveal the process of shifting into the +1 frame (Figure 1D). In the context of full-length mRNA-tRNA pairs that cause +1 frameshifting, both pairs move from the P site to occupy a position between the E and the P site (Figure 2C, 2D). This mRNA-tRNA placement is similar to the translocation intermediate containing EF-G (Zhou et al., 2013). In the structure presented containing an e*/E tRNA^Pro^ with the EF-G containing structures containing an pe/tRNA, the tRNA moves slightly away from the mRNA. In the e*/E tRNAPro CGG structure, the first position of the codon-anticodon forms a Watson-Crick interaction but the second and third nucleotides of the codon-anticodon are not within hydrogen bonding distances (Figure 2G). Interestingly, in the EF-G bound translocation intermediate structure, the codon-anticodon also is not within bonding distance (Zhou et al., 2013). These results suggest that it is the disruptive nature of moving between tRNA binding sites that perturbs the interactions between the codon and anticodon and these interactions likely reform once the transition to the next tRNA binding site is completed.”

If the interactions are different, does this diminish the relevance of the ASL-containing structures?

To recap, the ASL structures permit us to capture ribosome that have frameshifted into the new mRNA frame (Figure 1D). If we had used full-length tRNAs, the instability of the tRNA resulting from with the absence of methylation of G37 or the tRNA bound to a slippery codon, would have caused the tRNAs to not be gripped by the P site and move to the position between the E and the P site that we observe in Figures 2C,2D. Therefore, these structures are highly relevant because they demonstrate unequivocally that the ASL alone can cause the shift into the new frame.

4) The schematic model presented in Figure 5 is difficult to decipher. Steps are either missing or unnecessarily complicated. For example, why does the color of the 30S head domain change?

We have decided to remove the model Figure 5.

Reviewer #2:The paper presents several crystal structures of tRNA^Pro^ (ASL or full-length) on its cognate codon and a near-cognate codon in a slippery context that induces +1 frameshifting. The authors compare tRNAs with and without the modification m^1^G37 to understand the effect of the modification on frameshifting.1) The main solid conclusion of the paper is that even the modified tRNA^Pro^ can frameshift on a near-cognate/slippery codon. This is nice, but the conclusions come from the experiments with the ASLs.

This is not correct and we need to clarify these points. Yes we see that the tRNA is destabilized even when it is modified when bound to a near-cognate, slippery codon. These structures are not with ASLs though, but with full length tRNAs.

The question I have is whether this agrees with the previous functional analysis, i.e. what is the frameshifting frequency of tRNA^Pro^ on CCU/C in cells?

Yes it is well known that tRNA^Pro^ frameshifts in cells starting with the Bjork, Wikstrom and Bystrom, 1989 paper. Further, the Hou group measured frameshifting efficiencies using kinetic and reporter assays (Gamper et al., 2015). This is discussed in the second paragraph of the Introduction.

Are the ASL structures biologically relevant?

We argue that yes these structures are relevant given that they demonstrate that the ASL *alone* can cause the frameshift consistent with previous biochemical assays of other +1 frameshifting ASLs (Phelps et al., 2006; Walker and Frederick, 2006).

To recap, the ASL structures permit us to capture ribosome that have frameshifted into the new mRNA frame (Figure 1D). If we had used full-length tRNAs, the instability of the tRNA resulting from with the absence of methylation of G37 or the tRNA bound to a slippery codon, would have caused the tRNAs to not be gripped by the P site and move to the position between the E and the P site that we observe in structures shown in Figures 2C,2D. Therefore, these structures are highly relevant because they demonstrate unequivocally that the ASL alone can cause the shift into the new frame.

2) The authors present structures with ASLs and the full-length tRNAs. I am not sure I understand why show ASLs if similar information is available from the tRNAs. this has to be clearly described.

As noted above, these structures provide different insights: (1) The 70S-ASL structures demonstrate that frameshifting can occur in the P site and is influenced by the ASL of the tRNA alone. (2) The tRNA structures demonstrate that a near-cognate interaction between the codon-anticodon can cause destabilization in the P site that is similar in the context of a near-cognate interaction and tRNA^Pro^ lacking the m^1^G37 modification. We emphasized these points again in the Discussion.

3) I am not sure how relevant are the structures with a single deacylated tRNA in the P site or in the e*/E state. The authors indicate that this state represents ribosomes after peptide release. This is correct, but irrelevant for frameshifting.

We previously demonstrated that +1 frameshifting by tRNA^Pro^ does not occur in the A site (Maehigashi et al., 2014) and the Hou group demonstrated it can occur during translocation of the tRNA from the A to the P site or upon arrival in the P site (Gampet et al., 2015). We asked a simple question- how does ASL/tRNA^Pro^ interact with the P site since we know that this is where the shift into the new frame occurs? Given previous kinetic assays (Gampet et al., 2015) and other published biochemical assays showing that ASLs alone can cause a shift in the frame (Phelps et al., 2006; Walker and Frederick, 2006), and finally our resulting structures, we do not think it is irrelevant. This is explained in the first Results section and we emphasize these points in the Discussion shown below.

“This relative absence of interactions in the P and E sites provides an opportunity for the mRNA to shift out of frame. tRNA^Pro^ isoacceptors frameshift either during translocation from the A to the P site or after translocation in the P site (*11, 43*). Therefore we sought to capture the interactions of tRNA^Pro^ during a frameshift event. The structure of ASL^Pro^ bound to a slippery codon in the P site reveals the codon-anticodon interaction has shifted into the +1 frame indicating that the anticodon stem-loop interaction with the mRNA codon alone is important (Figure 1D). This frameshift is likely possible because interactions with the P-site tRNA are limited to only 16S rRNA nucleotides G1338 and A1339 (called the P/E loop nucleotides) with the anticodon stem and 16S rRNA C1400 with the anticodon nucleotide 34 (Figure 5A—figure supplement 5).”

The rotated swiveled state of the ribosome with the +1 frameshifting tRNA is very interesting, but what is the evidence that it occurs in cells?

It is known that the 30S head domain swivels and tilts during translocation of tRNAs from both prior structures and smFRET experiments (Wasserman et al., 2016; Ratje et al., 2010; Zhou et al., 2013, 2014). We observe this same swivel/tilt state in our structures. Since kinetic assays demonstrate that tRNA^Pro^ isoacceptors shift into the new frame during translocation or once the tRNA arrives in the P site (Gamper et al., 2015), this is strong evidence that the ribosome conformation we have trapped is a consequence of the mRNA-tRNA pairs. Further, we solved so-called “control” structures of these same tRNAs but in the context of mRNA-tRNA pairs that do not frameshift and these pairs do not elicit the same swivel/tilt of the 30S head domain.

We have expanded the Introduction, Results and Discussion sections at multiple places to ensure this is clear.

4) The model in Figure 5 is entirely speculative and misleading with respect to the fate of the E-site tRNA. In state 1 the E-site tRNA is indicated which then dissociates upon action of EF-G. Looking at the recent papers of Puglisi, Rodnina and Cooperman, the favored scenario, at least during normal translation, is that E-site tRNA dissociates before or simultaneously with the ternary complex binding. This should be changed in the model.

We have removed the model figure.

However, the Puglisi, Rodnina and Cooperman models are not mutually exclusive regarding E-tRNA dissociation and ternary complex delivery. Briefly, Cooperman has published previously that E-tRNA needs to dissociate before ternary complex binding however, this in the context of structured mRNAs (PMID 23542154). Rodnina has previously argued against this model (PMID 8901554) initially presented by Nierhaus as the allosteric three site model (PMIDs 22865895, 17013564). Data by the Green lab argue against this model (PMID 22378789).

Then, the +1 frameshifting pathway is entirely speculative. The structures in this paper do not have a tRNA in the E site – so why do the authors include it?

This is not correct. Although we sought to place the tRNA in the P site, in certain mRNA-tRNA contexts that cause frameshifting, the tRNA moves towards the E site. So in fact, we do have an E-site tRNA.

It is also not clear from the existing data at which step of translocation +1 frameshifting takes place and to the best of my knowledge there is no independent evidence in support the model.

Kinetic analyses of tRNA^Pro^ show that the lack of m^1^G37 causes frameshifting during both translocation of the tRNA from the A to the P site and when the tRNA is in the P site (Gamper et al., 2015). We have now expanded the Introduction to make this clear as shown below:

“However, this miscoding event does not cause the shift in the mRNA frame in the A site despite the tRNAPro CGG nucleotides of the anticodon stem-loop becoming more mobile (Maehigashi et al., 2014). Additional evidence that the frameshift event occurs in a post-decoding step includes detailed kinetic analyses (Gamper et al., 2015). The absence of the m^1^G37 modification in tRNA^Pro^ causes both a ~5% frameshifting frequency during translocation of the mRNA-tRNA pair from the A to the peptidyl (P) site and ~40% frameshifting frequency once the mRNA-tRNA pair has reached the P site. When tRNA^Pro^ lacks m^1^G37 and decodes a +1 slippery codon, both the process of translocation and the unique environment of the P site appear to contribute to the inability of the ribosome to maintain the mRNA frame.”

5) The conclusion as to the importance of the m^1^G37 modification is not clearly discussed.

In the Introduction, we discuss that the m^1^G37 modification stabilizes the loop important for decoding (our papers Maehigashi et al., 2014 and Nguyen, Hoffer and Dunham, 2019) but frameshifting does not occur in the A site. This is further confirmed by kinetic analyses of tRNA^Pro^ that shows that the lack of m^1^G37 causes frameshifting during both translocation of the tRNA from the A to the P site and when the tRNA is in the P site (Gamper et al., 2015).

We include a new Discussion section to emphasize the importance of m^1^G37 in stabilizing the tRNA in the P site (first Discussion paragraph with a few lines below that emphasize this point).

“Here, we elucidated the role of the m^1^G37 modification in tRNAPro CGG, which was known to be important in stabilizing stacking interactions with anticodon nucleotides (Maehigashi et al., 2014) and in the prevention of frameshifting (Bjork, Wikstrom and Bystrom, 1989; Hagervall et al., 1993). We find that this single methyl group influences the overall stability of tRNAPro CGG in an unexpected manner that causes large conformational changes between the tRNA and the 30S head domain, a domain known to move extensively during translocation of the tRNAs (Wasserman et al., 2016; Ratje et al., 2010; Zhou et al., 2013, 2014). However, the methylation *alone* does not stabilize tRNAPro CGG on the ribosome and, instead, its position is heavily influenced by interactions with a distinct slippery proline codon. Our structures of different mRNA- tRNAPro CGG pairs on the ribosome allows the first detailed mechanistic insight into how mRNA frame maintenance is regulated by both the m^1^G37 modification and the stability of mRNA-tRNA interaction.”

6) The paper is not well written. It is full of imprecise statements such as "programmed by tRNA" (the ribosome can be programmed by mRNA, but not by tRNA)

We have updated the manuscript to remove this.

or 5' anticodon stem (I don't know what this means).

This is a minor point of the paper so we have updated this section to be more general as shown below:

“However, this miscoding event does not cause the shift in the mRNA frame in the A site despite the tRNAPro CGG nucleotides of the anticodon stem-loop becoming more mobile.”

The Results include large parts of text than definitely belong to the Introduction, such as the beginning of section 1 and 3.

We have moved significant parts of the Results to the Introduction section.

Parts of the Results are speculations that may be more appropriate for the Discussion.

The original paper had a combined Results and Discussion section. We have now separated the Results and Discussion sections.

The text describing Figures 4 and 5 is very technical and very long and the conclusions are unclear.

We have separated the Results and Discussion sections and significantly updated the Results section.

Numbering of the supplemental figures does not correspond to the numbering in the text.

We have reviewed our figure callouts and all are correct. We would appreciate the reviewer directly stating which callouts are not correct.

Some of the supplemental figures would be better presented as main Figures, e.g. Figures 1 and 2 are not very informative, could be strengthened by combining with the respective supplementary figures.

We disagree that Figure 1 is not useful as the codon-anticodon interaction is an important observation to demonstrate that frameshifting has occurred. Figure 2 and S4 have been combined as per Reviewer 1 (see above).

The Abstract is difficult to understand due to the lack of precision.

The Abstract has been substantially rewritten. We are thankful for constructive criticism of our manuscript to help make the manuscript better. However without any specifics from the Reviewer, it is unclear what part of the Abstract lacks precision.

Reviewer #3:In this rich but dense paper, the authors discuss six crystal structures of ribosomes with ASL (2: cognate and slippery codons) or tRNA^Pro^ (4: cognate and slippery codons with and without modification at G37) programmed at the P site in order to study mechanistically frameshifting. This is remarkable. They show that the slippery codon is more critical for frameshifting than m^1^G37 modification.

We thank the reviewer for acknowledging the importance of our structures.

One clarification is that we show that a near-cognate, slippery codon elicits destabilization of the tRNA in the P site. With a slippery codon and a tRNA lacking m^1^G37, the tRNA is similarly destabilized causing the ASL, mRNA and the 30S head domain to be dynamic and uninterpretable in our structures.

This kind of article is extremely difficult to peer-review in the absence of the availability of, for example, Pymol sessions where one can visualize the whole regions of interest easily. We can only rely on the static figures selected by the authors (and without stereo views).

We uploaded all 6 PDBs and accompanying structure files for the reviewer to review in any program they choose, including PyMol, as required by *eLife* and other journals.

We have now made two videos to help in clarifying the changes we observe, which could be associated with the published work as supplementary materials in addition to their utility for the review process.

I would have liked to have the equivalent of Figure 1 for the six structures. This would have helped during the comparisons.

We do include these figures (Figure S4). As per reviewers 1 and 4, we have now combined Figures 2 and S4 (please see above).

In Figure 1D, U32 is positioned next to a G base, is this right?

U32 is adjacent to the U turn (U33; not labeled) and G31.

The size is also not the same as in B and C.

We thank the reviewer for this careful observation and have corrected this.

It would be nice to see better what C+1 is doing; no contact with G37?

C_+1_ does not interact with G37 and does not interact with tRNA or the ribosome. We have now clarified this point below:

“The first nucleotide of the proline codon, C+1, no longer interacts with the tRNA and does not appear to make any interactions with the ribosome.”

Since the authors used chemically synthetized ASL, it is surprising they did not look at crystal structures of ASL without G37 modification (no crystals maybe?).

Yes we typically attempt to solve structures of all different complexes but we could not obtain high-resolution structures of this particular complex.

In Supplemental Figure 3, why is the drawing for near cognate in the absence of G37 modification not shown (disorder?).

Yes the reviewer is correct- we did not include this particular complexes because, in this structure, i.e. the near-cognate interaction with tRNA^Pro^ lacking the m^1^G37 modification, there is a disordering of the ASL of the tRNA, the E-site mRNA codon and the 30S head domain (assessed by a reduction in the quality of the electron density) as we describe in the Results and Discussion section.

[Editors’ note: what follows is the authors’ response to the second round of review.]

Revisions:1) The physiological relevance of the findings, given the use of deacylated tRNA in the P site and a very short mRNA used for the experiments in the paper, is still not clear.

As we noted in the previous response, there are dozens of ribosome structures with deacylated tRNAs bound at the P site. In these structures, the tRNAs adopt normal positions in the P site of the ribosome and have never been observed to cause conformational rearrangements like those in our structures presented in this manuscript. This is because these tRNAs, unlike those we examine here, do not cause mRNA frameshifts; the differences we observe in our structures therefore have nothing to do with the acylation state of the tRNAs used. (Furthermore, as noted previously, had we used aminoacylated tRNA^Pro^, the tRNA would have become deacylated within the lifetime of the experiment, adding a significant practical challenge to addressing a concern about a feature of our structures that has no bearing on their physiological relevance.)

As for the use of the short mRNA, this was necessary to define the mRNA frame by using different lengths of mRNA. We discuss the reasons for why we used this mRNA in the manuscript at the two places noted below:

“The mRNA used in these studies only contained a single nucleotide after the three-nucleotide codon programmed in the P site to allow for the unambiguous identification of the reading frame.”

“Similar to the structural studies of the ASL^Pro^ in the 0 frame (Figures 1B and D), the mRNA used in these studies contains an additional nucleotide after the P-site CCG codon (Figures 1; Figure 2B and Table 1). The mRNA has clear phosphate density for the single A-site nucleotide indicating the mRNA remains in the 0 frame (Figure 2—figure supplement 1D).”

The authors argue that +1 frameshifting is promoted by the tRNA itself, which justifies the use of the minimal model system, and state that the presence of an A-site tRNA and the nascent peptide do not matter. The authors cite Phelps et al., 2006 and Walker and Fredrick, 2006 as support for this statement. However, these papers, and the previous work of the authors, are in vitro studies carried out with deacylated tRNAs and there is no systematic analysis of whether or not the A-site tRNA and nascent peptide affect the frameshifting efficiency. The work by Björk, Wikstrom and Bystrom, 1989 has been carried out in the background of the SufA6 suppressor, rather than with a native tRNA(Pro).

What we previously argued was that it is not such a far-fetched idea that tRNAs alone can promote frameshifting as evidenced by dozens of frameshift suppressor tRNAs which have been known for decades to cause mRNA frameshifting.

We agree there is no systematic analysis of every detail of the tRNA we used in our studies. But the kinetic analyses by the Hou group of tRNA^Pro^ support the idea that it is the tRNA alone that can cause frameshifts. Our previous structures of tRNA^SufA6^ bound at the P site (Hong et al., 2018) are also strong support that tRNA^Pro^ could function similarly to tRNA^SufA6^ (given that tRNA^SufA6^ is derived from tRNA^Pro^). Finally, if the nascent chain or the A-site tRNA *did* influence the ability of tRNA^Pro^ lacking the m^1^G37 to frameshift, we would never have seen +1 frameshifting in our structures, but we do.

In light of these concerns, we have updated the following sentence in the Introduction:

“Furthermore, the presence of a nascent polypeptide chain, the acylation status of the tRNA (deacylated or aminoacylated), and even the presence of a tRNA in the A site do not appear to contribute to the ability of these tRNAs to cause frameshifting.”

The study by Bjork in 1989 was the first to show the importance of the m^1^G37 modification in tRNA^Pro^. These authors used different strains to demonstrate this but follow up genetic experiments by the Bjork and Farabaugh groups provided more support for the importance of m^1^G37.

In contrast, the paper by Gamper et al., 2015, does not support the statements of the authors. Gamper et al. show that significant +1 frameshifting on a CCCC sequence (which is likely more "slippery" than the CCCU sequence used here) not only depends on the absence of the m^1^G37 modification, but also on the absence of EF-P and the position of the slippery sequence in the gene. Even in the absence of the m^1^G37 modification, the frameshifting efficiency is very low (<2%) in the presence of EF-P and in the middle of a gene (i.e. at physiological conditions), and is only stimulated at particular conditions, e.g. when the slippery codon is engineered at position 2 of the gene or is followed by a rare codon (Figure 1 in the Gamper et al. paper). Thus, although the CCC C/U sequence can induce frameshifting in the absence of the m^1^G37 modification, the efficiency obviously depends on the nascent peptide (middle gene positions vs. position 2) and the presence of the A-site tRNA (demonstrated by the rare codon effects in the A site). The authors should clearly acknowledge these previous findings.

As we describe in the Introduction, prior genetic experiments show that tRNA^Pro^ can frameshift on either CCC-U or CCC-C codons (Sroga et al., 1992; Qian et al., 1998). Gamper et al. used the CCC-C codon for their kinetic experiments and never tested CCC-U. Therefore, it cannot be assumed that one codon is more “slippery” than the other as the reviewer suggests.

Additionally, Gamper et al. show that EF-P can suppress the +1 frameshift and the location of the slippery sequence on the gene may also influence frameshifting. We do agree that EF-P influences +1 frameshifting exhibited by tRNA^Pro^ and already discuss this in the Discussion section. We have also now included an additional statement to mention the possible influence of the location on the mRNA, nascent chain lengths and rare codons on frameshifting in the Discussion:

“An additional difference is the dependency on elongation factor-P (EF-P) to reconcile +1 frameshifts. EF-P binds at the E site to overcome ribosome stalling induced by poly-proline codons (Use et al., 2013; Huter et al., 2017) and is critical in suppressing frameshifts on poly-proline stretches (Gamper et al., 2015). While EFP reduces +1 frameshifts with tRNAPro CGG G37 (Dm^1^) to an equivalent frequency as native tRNAPro CGG, there is little influence on isoacceptor tRNAPro cmo/UGG in the absence or presence of m^1^G37.”

“…These mechanistic differences may be due to a combination of the codon-anticodon pairings on slippery codons (tRNAPro cmo/UGG and tRNAPro CGG are cognate with slippery codons while tRNAPro CGG is near-cognate (Nasvall, Chen and Bjork, 2004)) and/or the influence of other modifications such as the cmo^5^-U34 modification in tRNAPro cmo/UGG (Masuda et al., 2018). Further considerations may be the location of the slippery codon on the mRNA, which would have varying nascent chain lengths, and whether the following codon after the slippery codon is rare.”

The issue with a very short mRNA is even more serious, because a short mRNA may be much more dynamic than that of a longer construct and adopt conformations that are not populated by a longer mRNA.

We respectfully disagree with this assertion. First, we determined two structures of tRNA^Pro^ at the P site with this same short mRNA and do not observe these conformational changes (Figures 2A,B). These controls, which do not undergo frameshifting, provide conclusive evidence that the 30S head rotation we observe in other structures (Figure 2C) has nothing to do with the mRNA length. Second, more generally, inducing dynamic features in a complex as the reviewer suggests we may have done could result in three most likely outcomes in a crystallographic experiment: (1) no crystals, (2) crystals that diffract more poorly, or (3) a lack of density for the dynamic feature. None of these are the case here as shown in (Figures 1E, F, G and Figure 2—figure supplement 1), with well-defined mRNA electron density observed.

2) Revision to the Abstract are needed. First sentence should probably read "Modifications in the tRNA adjacent to the three-nucleotide anticodon".

We have updated the text to the following:

“Modifications in the tRNA anticodon loop, adjacent to the three-nucleotide anticodon, influence…”

The sentence "by stabilizing the tRNA to allow for accurate reading…" is misplaced, because this is a result of the work, which is indicated in the fourth sentence (…destabilizes interactions…).

It has been known from many studies that modification of the nucleotides outside the anticodon, but in the anticodon loop and especially at nucleotide 37, contribute to the stability of the reading of the mRNA frame. This is the point we are making and we are not referring to our work.

Replace "peptidyl site" with the "P-site of the ribosome".

We have updated this to: “P site of the ribosome”.

However, because the “P site” is being used as a noun and not an adjective, it should not be hyphenated.

Remove or rephrase the part of the final sentence, as it is not reflecting the results of the work in an adequate way (the slippery codons are known to affect frameshifting).

The reviewer is referring to the last sentence of our Abstract (shown below):

“These studies provide molecular insights into how m^1^G37 stabilizes the interactions of tRNA^Pro^ with the ribosome and the influence of slippery codons on the mRNA reading frame.”

Our structures do provide these insights.

3) Although the CCCU codon can promote frameshifting in some contexts it is not slippery by itself. The recommendation is to better explain this in the Introduction and use "CCCU codon" throughout the text, rather than "near-cognate, slippery codon".

We have updated the way we call this out throughout the text. We have additionally removed the word “near-cognate” in the figure legends.